



# A Monte Carlo approach for determining cluster evaporation rates from concentration measurements

Oona Kupiainen-Määttä[1]

[1]University of Helsinki, Department of Physics, P.O. Box 64, FI-00014 University of Helsinki, Finland

*Correspondence to:* Oona Kupiainen-Määttä (oona.kupiainen@alumni.helsinki.fi)

**Abstract.** Evaporation rates of small negatively charged sulfuric acid–ammonia clusters are determined by combining detailed cluster formation simulations with cluster distributions measured at CLOUD. The analysis is performed by varying the evaporation rates with Markov chain Monte Carlo (MCMC), running cluster formation simulations with each new set of evaporation rates and comparing the obtained cluster distributions to the measurements. In a second set of simulations, the fragmentation of clusters in the mass spectrometer due to energetic collisions is studied by treating also the fragmentation probabilities as unknown parameters and varying them with MCMC. This second set of simulations results in a better fit to the experimental data, suggesting that a large fraction of the observed $HSO_4^-$ and $HSO_4^- \cdot H_2SO_4$ signals may result from fragmentation of larger clusters, most importantly the $HSO_4^- \cdot (H_2SO_4)_2$ trimer.

## 1 Introduction

Gas-phase sulfuric acid has long been believed to be an important precursor for particle formation in the atmosphere (Doyle, 1961; Kiang et al., 1973; Cox, 1973; Mirabel and Katz, 1974). The details of the process have, however, remained poorly understood until lately. Recent laboratory experiments (Berndt et al., 2010; Sipilä et al., 2010; Benson et al., 2011; Almeida et al., 2013) have confirmed that particle formation rates of the magnitude observed in the atmosphere can be produced with ambient sulfuric acid concentrations and either impurities present in laboratory air or intentionally added low concentrations of base molecules, giving new support for sulfuric acid being at least one of the compounds driving atmospheric particle formation.

New high-resolution mass spectrometers have enabled the detection and characterization of individual ionic clusters consisting of only a few molecules (Junninen et al., 2010), opening a new window into the first steps of cluster formation. However, measurements alone cannot fully uncover the dynamics of the process, as they only provide information on the concentrations, not the collision and evaporation fluxes from one cluster type to another.

At the same time, modeling of particle formation has also advanced greatly in the past few years. For the first time theoretical predictions of cluster distributions (Olenius et al., 2013b) and particle formation rates (Almeida et al., 2013) agree qualitatively with experimental findings.

Cluster formation simulations require as input the collision and evaporation rates of clusters. The collision frequencies are usually computed simply using classical physics, and an estimate of the evaporation rates can be obtained by relying on



equilibrium considerations and using the formation free energies of the clusters computed by quantum chemistry. This approach has been shown to give qualitative agreement with experiments (Almeida et al., 2013; Olenius et al., 2013b), but several very drastic assumptions are involved. First-principles molecular dynamics simulations (Loukonen et al., 2014a, b) have shown that one harmonically oscillating cluster structure is far from a realistic description of the thermal motion of molecules in a cluster,

implying that the traditional way of computing cluster formation free energies may be a rough approximation. As evaporation rates depend exponentially on the cluster formation energies, theoretical evaporation rates may easily be wrong by several orders of magnitude. Also the treatment of the collision rates is highly simplified, but errors of more than a factor of two or perhaps ten are unlikely.

An alternative approach for estimating the rate constants is to start from experimental cluster concentrations and find rate

constants that reproduce these results. This has been done previously by Bzdek et al. (2010) who measured time series of cluster concentrations in order to study base exchange in positively charged clusters containing a fixed number of sulfuric acid molecules, and by Jen et al. (2014) who measured concentrations of neutral clusters containing two sulfuric acid molecules in the presence of different base compounds. However, in both cases the studied system consisted of only a few cluster types, and the theoretical description was highly simplified. Bzdek et al. (2010) assumed sequential pseudo–first order substitution

reactions, and used the analytic solution of the time evolution of the concentrations to fit the pseudo–first order rate constants. Jen et al. (2014), on the other hand, used a heuristic cluster formation model with only two free parameters to optimize. In both cases, the optimization problem was simple enough that traditional fitting tools could be used. More recently, Chen et al. (2015) used a more complicated model with tens of unknown parameters to describe measured particle concentrations in an experiment involving methanesulfonic acid, trimethylamine and water, but they used effective reaction rates instead of separate

collision and evaporation rates, and only presented one reasonably good fit instead of attempting to find either the best fit or all sets of parameter values giving a good fit.

In this study, measured cluster distributions are combined with detailed cluster formation simulations describing explicitly all possible collision and evaporation processes. Theoretical estimates are used for the collision rates, while all evaporation rate coefficients as well as some parameters related to experimental details are optimized to reproduce the experimental data.

Due to the large number of unknown parameters, the fitting is done by Monte Carlo simulation. The method is applied to measurement data from the CLOUD experiment (Olenius et al., 2013b). This study focuses solely on ion clusters, but a similar approach could also be used for determining evaporation rates of neutral clusters based on cluster distributions measured with a chemical ionization mass spectrometer.

## 2    Experimental ion cluster distributions

The experimental cluster distributions used in this study are from an earlier publication from the CLOUD experiment at CERN (Olenius et al., 2013b). Concentrations of negatively charged sulfuric acid–ammonia clusters were measured in steady-state conditions with sulfuric acid vapor concentrations between $10^7$ and $10^9$ cm$^{-3}$ and ammonia mixing ratios from below 35 ppt up to 250 ppt. The clusters were detected using a high resolution APi-TOF (Atmospheric Pressure interface Time-Of-Flight)



mass spectrometer. The largest clusters considered in the study contained one $HSO_4^-$ ion, four $H_2SO_4$ molecules and four ammonia molecules. However, it is likely that most of the clusters initially also contained some water molecules, although none were detected, and water was concluded to evaporate from the clusters inside the APi-TOF. The clusters were also assumed to lose some or all of the ammonia molecules inside the instrument prior to detection. Therefore, the concentrations were

reported separately for ammonia containing and ammonia-free clusters, but the ammonia containing ones were not sorted further by number of ammonia molecules. The bisulfate ion $HSO_4^-$ and the two smallest clusters, $HSO_4^- \cdot (H_2SO_4)_{1-2}$, were only observed with no ammonia molecules attached. Olenius et al. (2013b) presented a total of 25 cluster distributions measured with ion production from natural ionization, a temperature of 278 K and different sulfuric acid and ammonia vapor concentrations, but three of these distributions had very low concentrations for some of the cluster types and were thus omitted

from the present study.

## 3 Simulation methods

Cluster dynamics simulations were performed with ACDC (Atmospheric Cluster Dynamics Code), a program that writes out the birth-death equations for a given set of molecules and clusters and solves them by numerical integration. Unlike in earlier implementations of ACDC where MATLAB was used, the birth-death equations were now integrated using the Fortran ordinary

differential equation solver VODE (Brown et al., 1989). A detailed description of the code has been published elsewhere (McGrath et al., 2012; Olenius et al., 2013a), and only the main points and the differences to the earlier version are presented here.

### 3.1 ACDC simulations

To minimize the computational burden of solving the birth-death equations, only negatively charged clusters were consid-

ered. Both quantum chemical calculations and mass spectrometry measurements indicate that negatively charged clusters with three sulfuric acid molecules or less (including the bisulfate ion) do not take up ammonia molecules (Kirkby et al., 2011; Olenius et al., 2013b). Based on the main formation pathway in cluster formation simulations (Olenius et al., 2013a), the clusters $HSO_4^- \cdot (H_2SO_4)_{0-2}$, $HSO_4^- \cdot (H_2SO_4)_3 \cdot (NH_3)_{0-3}$ and $HSO_4^- \cdot (H_2SO_4)_4 \cdot (NH_3)_{0-4}$ were chosen to form the simulated system in this study. The only electrically neutral species included in the simulation were the $H_2SO_4$ and $NH_3$

monomers.

Some of the negatively charged clusters could in principle result from collisions of neutral clusters with negative ions, but both experimental observations (Jen et al., 2014) and quantum chemical calculations (Olenius et al., 2013a) suggest that sulfuric acid–ammonia clusters are so weakly bound that their concentrations are orders of magnitude lower than the sulfuric acid monomer concentration at conditions corresponding to the experiments reported by Olenius et al. (2013b). Therefore, the

contribution of neutral clusters was not taken into account in this study. Water molecules were not modeled explicitly, but the collision and evaporation coefficients should be interpreted as effective rates averaged over the hydrate distribution of each cluster type (see for instance Paasonen et al., 2012).





In addition to growing by collisions with monomers or decaying by monomer evaporations, the negative clusters can get neutralized by recombination with positively charged ions and clusters. To keep the situation simple, the distribution of positive clusters was not simulated explicitly, but the overall positive ion concentration was set to match the total negative ion concentration, and all negative ions were assumed to have the same recombination rate coefficient of $1.6 \times 10^{-6}$ cm$^3$s$^{-1}$ (Israël, 1970)

with these generic positive ions. The formed neutral clusters were outside the system of interest, and their concentrations were not recorded.

The formation of negative ions was modeled similarly as was done by Almeida et al. (2013). Generic charger ions with the properties of $O_2{}^-$ are first produced at a constant rate, and upon collisions with $H_2SO_4$ molecules they ionize these to form bisulfate ions. The charger ions can also be lost by recombination with positive ions. Finally, all clusters and charger ions can

be lost on the chamber walls, and this was described by a size- and composition-independent wall loss coefficient.

To mimic the experimental conditions as closely as possible, each simulation was started from a situation with non-zero sulfuric acid and ammonia monomer concentrations and no ions. The charger ion source was switched on, and the time evolution of the cluster concentrations was simulated keeping the neutral monomer concentrations constant. The experimental cluster distributions correspond to steady-state conditions (Olenius et al., 2013b), and the lengths of the individual experiments were

of the order of half an hour (Kirkby et al., 2011). The modeled cluster distribution was calculated as an average of the distributions at time $t_1 = 20$ min and $t_2 = 30$ min after the beginning of the run. The extent to which the simulation had reached a steady state was characterized by the ratio of the concentrations at $t_2$ and $t_1$, calculated in each case for the cluster for which this ratio deviated most from unity. This convergence parameter was used together with the cluster concentrations to determine how well the simulations reproduced the experimental results.

## 20  3.2   Simulation parameters

As the measurement data consisted of steady-state concentrations, it was not possible to fit both the collision and evaporation rates – multiplying all rate constants by the same factor would only change the timescale of the process but not the steady-state concentrations. Collision frequencies between ions and polar or polarizable molecules can be approached theoretically by considering classical electrostatic interactions. While a closed-form analytical expression cannot be obtained even when

neglecting quantum effects, theoretical estimates for collision rates are much more reliable than those for evaporation rates. In all the simulations presented in this study, the collision rate constants were computed using the parameterization of Su and Chesnavich (1982) based on classical trajectory simulations. The values for the reactions in the studied system were between $10^{-9}$ and $4 \times 10^{-9}$ cm$^3$s$^{-1}$.

In principle, the evaporation rates might have any values, and there is no way to constrain even their order of magnitude

based on earlier experimental evidence or simple theoretical considerations. However, the interval in which the evaporation rates are allowed to vary does not in practice need to be infinitely wide. If the length of the simulation is 30 minutes, it does not matter whether a cluster has a lifetime of one day or one week – it will in any case not evaporate. On the other hand, if a cluster collides with monomers on average once per second or once per minute, there is no effective difference whether it has an evaporation lifetime of one millisecond or one microsecond – it will almost certainly evaporate before it has a chance



to grow further. Even so, the range of interest for the evaporation rates spans several orders of magnitude, and the base ten logarithms of the rates (used as the parameters to be varied by MCMC instead of the rates themselves) were sampled from the range of -10 to 10.

The simulations also involve a large number of experiment-related parameters whose values cannot be measured directly or estimated reliably based on any fundamental theory. These were also treated as free parameters and varied using MCMC. For some of the parameters, however, at least an order-of-magnitude estimate is available, and these estimates were used for constraining the range in which the parameters were allowed to vary.

A wall loss rate of $1.7 \times 10^{-3}$ s$^{-1}$ was determined for the electrically neutral $H_2SO_4$ monomer in the CLOUD chamber (Almeida et al., 2013). This rate decreases with increasing cluster size, but ions may have a higher loss rate. For simplicity, all clusters were assumed to have the same wall loss rate, and its value was sampled from the range 0 and $10^{-2}$ s$^{-1}$. Based on measured ion concentrations and approximate loss rates of ions, the ion production rate due to natural ionization was estimated to be of the order of 3 ion pairs cm$^{-3}$s$^{-1}$ (Olenius et al., 2013b). In this study, it was sampled from the range of 0 to 10 ion pairs cm$^{-3}$s$^{-1}$.

In some experiments, no ammonia was added intentionally to the chamber. While its concentration was in these cases below the detection limit of 35 ppt, some trace amount must have been present as ammonia molecules were observed in the clusters. In the simulations, two approaches were used regarding the ammonia concentration: either a constant background ammonia mixing ratio of 5 ppt was used for all these experiments, or the mixing ratio was allowed to vary separately for each of these low-ammonia experiments, and the values were sampled between 0 and 50 ppt.

### 3.2.1 Fragmentation in the mass spectrometer

It is possible that some clusters fragment inside the instrument before detection. Weakly bound water molecules probably evaporate to a great extent (Ehn et al., 2011), and they are not taken explicitly into account in the cluster distribution. Also ammonia and sulfuric acid molecules may be detached from the clusters due to energetic collisions with gas molecules when the clusters are accelerated inside the instrument. In some of the MCMC simulations, all clusters were allowed to fragment, and the fragmentation probabilities were sampled between 0 and 1, with the constraint that the sum of all fragmentation probabilities corresponding to the same cluster fragmenting to form different products could not be higher than one.

In an IMS-TOF (ion mobility spectrometer – time-of-flight mass spectrometer) experiment, detachment of sulfuric acid molecules was observed to be important at least for the pure trimers, $HSO_4^- \cdot (H_2SO_4)_2$, which can lose either one or two $H_2SO_4$ molecules (Adamov et al., 2013). In the present study, each of the pure sulfuric acid clusters $HSO_4^- \cdot (H_2SO_4)_i$ could fragment through $i$ different processes with separate fragmentation probabilities, forming the products $HSO_4^- \cdot (H_2SO_4)_{0,1,2,\ldots,(i-1)}$.

On the other hand, in another IMS-TOF experiment, larger sulfuric acid–dimethylamine clusters $HSO_4^- \cdot (H_2SO_4)_i \cdot ((CH_3)_2NH)_i$ with $i = 3, 4, 5$ were observed not to fragment (Bianchi et al., 2014). The fragmentation patterns of larger clusters containing sulfuric acid and ammonia have not been determined experimentally, and it is possible that fragmentation is more important than for the above-mentioned dimethylamine-containing clusters. However,




the larger the cluster, the more vibrational degrees there are to absorb any excess energy released in collisions, so the fragmentation probabilities can be expected to decrease with increasing cluster size Kurtén et al. (2010). For simplicity, detachment of sulfuric acid molecules from ammonia-containing clusters was not taken into account, although it might in reality occur to some extent, and the removal of ammonia molecules from the clusters was described by only four parameters:

the probabilities of detecting $HSO_4^- \cdot (H_2SO_4)_3 \cdot NH_3$ and $HSO_4^- \cdot (H_2SO_4)_3 \cdot (NH_3)_{2-3}$ clusters as pure acid tetramers and of detecting $HSO_4^- \cdot (H_2SO_4)_4 \cdot NH_3$ and $HSO_4^- \cdot (H_2SO_4)_4 \cdot (NH_3)_{2-4}$ clusters as pure acid pentamers. This choice of fragmentation-related parameters is a trade-off between describing the processes as accurately as possible and keeping the number of free parameters reasonable.

### 3.3 Monte Carlo simulations

The effect of the above-mentioned unknown parameters (evaporation rates, ion production rate, wall loss coefficient, background ammonia concentrations, fragmentation probabilities) on the cluster distribution was studied by Bayesian analysis using Markov chain Monte Carlo (MCMC). (See e.g. Brooks et al., 2011, for an introduction to MCMC methods.) The aim of MCMC in parameter estimation is to find combinations of parameter values that reproduce the experimental data as well as possible. Instead of finding one best fit, the objective is to find a distribution of the most likely parameter values. This is

accomplished by forming a chain $Z$ of parameter values that converges toward the desired distribution as the length of the chain increases.

### 3.3.1 The Metropolis algorithm

The parameters are varied using a random-walk approach, and at each step the new parameter values (denoted as the vector $\boldsymbol{x}_{new}$ with length $n_{coefs}$) are used for running ACDC simulations corresponding to all experiments. In the Metropolis algorithm,

the proposal density $q(\boldsymbol{x}_{new}, \boldsymbol{x}_{old})$ describing the probability of attempting a step from the old point $\boldsymbol{x}_{old}$ to a new point $\boldsymbol{x}_{new}$ is equal to the proposal density $q(\boldsymbol{x}_{old}, \boldsymbol{x}_{new})$ related to the reverse step from $\boldsymbol{x}_{new}$ to $\boldsymbol{x}_{old}$. The difference between the modeled and measured cluster distributions is quantified by the square sum

$$SS_{new} = \sum_{i=1}^{n_{out}} (\log_{10} y_{exp,i} - \log_{10} y_{new,i})^2 \tag{1}$$

where $n_{out} = n_e \times (n_c + 1)$ is the number of output values, $n_e = 22$ is the number of experiments, $n_c = 7$ is the number of cluster

types whose concentrations are measured, $\boldsymbol{y}_{new}$ is a vector of length $n_{out}$ containing simulated cluster concentrations for all runs as well as one convergence parameter (see Sect. 3.1) for each run, and $\boldsymbol{y}_{exp}$ is the corresponding vector for the experimental data with a value of 1 for the convergence parameter for all runs. The reason for including the convergence parameter here is to penalize low wall loss rates and ion source rates that would lead to an unrealistically slow time evolution of the cluster distribution.

Assuming that the experimental data contains measurement errors that can be described as uncorrelated multiplicative log-normal noise with the same variance $\sigma^2$ for each measured value $y_{exp,i}$, the likelihood of observing the data $\boldsymbol{y}_{exp}$ given the



parameter values $x_{\text{new}}$ is

$$p(y_{\text{exp}} \mid x_{\text{new}}) = \frac{1}{(2\pi\sigma^2)^{n_{\text{out}}/2}} \exp\left(-\frac{1}{2\sigma^2} SS_{\text{new}}\right). \tag{2}$$

At each step of the random walk, the value $SS_{\text{new}}$ is compared to the square sum $SS_{\text{old}}$ saved at the previous step. If the new value is lower or equal to $SS_{\text{old}}$, that is if the new parameter values reproduce the experimental data at least as well as the previous ones, the point is accepted. If, on the other hand, $SS_{\text{new}} > SS_{\text{old}}$, the point may still get accepted, but only with probability

$$\frac{p(y_{\text{exp}} \mid x_{\text{new}})}{p(y_{\text{exp}} \mid x_{\text{old}})} = \exp\left[-\frac{1}{2\sigma^2}(SS_{\text{new}} - SS_{\text{old}})\right]. \tag{3}$$

The overall acceptance probability for both cases can then be written as $\alpha = \min\left(1, \exp\left[-\frac{1}{2}\sigma^{-2}(SS_{\text{new}} - SS_{\text{old}})\right]\right)$. If the new point is accepted, the parameter values $x_{\text{new}}$ are saved to the chain $Z$ and $SS_{\text{old}}$ is replaced by $SS_{\text{new}}$. Otherwise the previous point $x_{\text{old}}$ is added again to the chain $Z$.

### 3.3.2   DE-MC$_Z$ algorithm for finding all local maxima of the distribution

Some parameters were found to have posterior distributions with more than one local maximum. Plotting two-dimensional posterior distributions of pairs of parameters showed in many cases L-shaped or otherwise non-convex regions of high probability that are hard to sample using traditional methods. In order to ensure that the random walk was able to find all the local maxima and converged to the correct distribution, the DE-MC$_Z$ algorithm (Differential Evolution Markov Chain algorithm sampling the difference vectors from the past) introduced by ter Braak and Vrugt (2008) was employed. In DE-MC$_Z$, several chains are run in parallel, and each chain in turn takes a step $x_{\text{new}} = x_{\text{old}} + \gamma(x_1 - x_2) + \delta$, where $\gamma$ is a scalar, $x_1$ and $x_2$ are two different randomly selected points from the joint history of all chains, $Z$, and $\delta$ is a small additional term drawn from a normal distribution with a small variance compared to the width of the posterior distribution. Ter Braak and Vrugt (2008) found that three chains worked well in their test systems, but in this study, five chains were used as they were noted to ensure better mixing. Based on the recommendations of ter Braak (2006) and ter Braak and Vrugt (2008) and on test simulations, $\gamma$ was set to 0.98 at every fifth step and $2.38/\sqrt{2 \times n_{\text{coefs}}}$ otherwise. The width of the distribution for sampling $\delta$ was based on an estimate of the width of the posterior distribution as discussed in the Supplementary Material. As the rule for proposing steps is symmetric with respect to $x_{\text{new}}$ and $x_{\text{old}}$ but depends on the history, the DE-MC$_Z$ algorithm is an adaptive Metropolis algorithm and the acceptance probability is calculated like in the basic Metropolis algorithm.

Further details about the MCMC simulations are presented in the Supplementary Material.

### 3.4   Overview of the simulations

An overview of the simulation methods is presented in Fig. 1. The same MCMC procedure (shown in orange in the Figure) was used with two alternative sets of cluster distributions as input. In both cases, these cluster distributions corresponded to 22 individual experiments (or computer experiments) with varying sulfuric acid and ammonia vapor concentrations, and for each



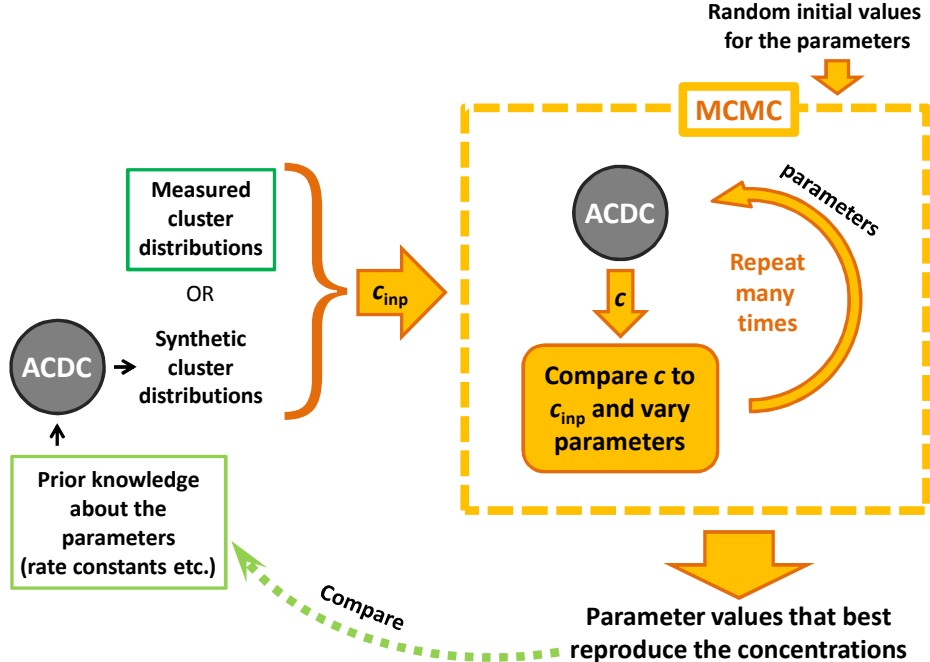

**Figure 1.** Schematic representation of the steps involved in the study. The green boxes show the two alternative starting points.

experiment the concentrations of seven cluster types were included in the distribution. In the MCMC simulation, all unknown parameters (evaporation rates etc.) were first given some random values, and these were used for running a set of 22 ACDC simulations with vapor concentrations corresponding to the input cluster distributions. The cluster concentrations obtained from the ACDC runs were compared with the input cluster concentrations, and the parameters were given new values. The new parameter values were again used to run a set of ACDC simulations, and the process was repeated over and over.

The starting point of the main part of the study (dark green box) were the 22 cluster distributions measured at CLOUD at varying sulfuric acid and ammonia vapor concentrations. These were used as input for an MCMC simulation, and the main output of the MCMC simulation were parameter values that reproduced most closely the measured cluster distributions. However, unlike traditional fitting procedures, MCMC gives a distribution of most likely parameter values (called the posterior distribution) and corresponding cluster distributions instead of one best fit.

The second part of the study focused on testing the performance of the MCMC data analysis method. First one possible set of parameter values was selected (light green box in Fig. 1). Quantum-chemistry based theoretical predictions were used for the cluster evaporation rates, and the other parameter values were estimated based on the experiment. These parameter values (referred to later in the paper as input parameter values) were used as input for a set of 22 ACDC simulations corresponding to the same sulfuric acid and ammonia vapor concentrations as in the experimental cluster distributions. The chosen input values of the fragmentation parameters were applied to the output concentrations from these ACDC runs to get a set of 22 cluster distributions. Some random noise (see Sect. 4.1) was added to these simulated cluster distributions to obtain synthetic





'measured' cluster distributions. These, in turn, were then used as input for an MCMC simulation, and the output was again a distribution of most likely parameter values as well as corresponding cluster distributions. Since in this case the 'correct answers', that is the input parameter values used to produce the synthetic cluster distribution, were known, the parameter distributions obtained as output from MCMC could be compared to the input values.

## 4 Results

Although the main result from the MCMC simulation are the distributions of likely parameter values, it is useful first to look at the cluster distributions corresponding to these output parameter values (referred to later as output cluster distributions) and check how accurately the input data is reproduced. Such comparisons are presented in Sect. 4.1 for the CLOUD data and two sets of MCMC simulations with a different set of free parameters. If the output cluster distributions are very far from the

measured cluster distributions, it can be concluded that the model used in the simulations did not correspond closely enough to the actual processes determining the observed cluster distributions. In such a case, the fitted parameters do not necessarily correspond directly to the corresponding real parameters, or indeed have any clear physical interpretation.

The output values of the evaporation rates and fragmentation probabilities are discussed in detail in Sects. 4.2 and 4.3, respectively, only for cases where the output cluster distributions reproduce closely the measured concentrations. The results

for the other parameters are presented in Sect. S3 of the Supplementary Material.

Even when the MCMC simulation finds a good fit to the observed distributions, the interpretation of the output parameter distributions is not always clear. To get better insight into what conclusions can safely be drawn, Sect. S2 of the Supplementary Material presents test simulations for synthetic input cluster distributions with known evaporation rates and fragmentation probabilities.

### 4.1 Cluster distributions

Figure 2 presents the experimental cluster distributions from CLOUD together with the output cluster distributions from an MCMC simulation where only the evaporation rates are varied and fragmentation in the mass spectrometer is not taken into account. The background ammonia concentration is set to 5 ppt, and the values reported by Olenius et al. (2013b) are used for the ion production rate and wall losses. The medians of each concentration from the output of the MCMC simulations

are presented as a horizontal line, and the vertical lines span between the 2.5th and 97.5th percentiles. Comparison of the measured and simulated concentrations shows that while overall the simulated concentrations are mostly of a correct order of magnitude, the MCMC fitting does not produce the correct precursor concentration dependence for all ion cluster types. In case of the bisulfate ion $HSO_4^-$ and the charged dimer $HSO_4^- \cdot H_2SO_4$, the measured ion concentrations are notably lower in the experiments with a high ammonia concentration than in experiments with a similar acid concentration and no added ammonia,

while the simulated concentrations show practically no ammonia dependence. For the larger clusters, on the other hand, the ammonia dependence is captured reasonably well. However, the sulfuric acid concentration dependence of the output cluster





**Figure 2.** Cluster distributions measured at CLOUD and the corresponding modeled cluster concentrations from an MCMC simulation where only the evaporation rates are varied and no fragmentation is allowed. A stands for $H_2SO_4$, $A^-$ for $HSO_4^-$ and N for $NH_3$.

distributions differs from the observed dependence also for many of the larger clusters at low ammonia concentrations. This discrepancy is most prominent for $HSO_4^- \cdot (H_2SO_4)_4 \cdot NH_3$ and $HSO_4^-$.

Using the ion production rate and wall loss constant as free parameters while still keeping a fixed background ammonia concentration does little to improve the fit. The same discrepancies remain also if the background ammonia concentrations are
5 varied.

Figure 3 presents the output cluster distributions from an MCMC simulation where the fragmentation probabilities discussed in Sect. 3.2.1 are treated as free parameters. The ion production rate and wall loss constant are also varied, but all background ammonia concentrations are set to 5 ppt. Apart from a few outliers in the experimental concentrations, the agreement between the measured and modeled concentrations is remarkably good. This suggests that the poor fit in Fig. 2 may be explained by the
10 concentrations observed by the mass spectrometer not corresponding directly to the ion concentrations in the CLOUD chamber,




**Figure 3.** Cluster distributions measured at CLOUD and the corresponding modeled cluster concentrations from an MCMC simulation where evaporation rates, fragmentation probabilities, the ion production rate and the wall loss rate are varied. A stands for $H_2SO_4$, $A^-$ for $HSO_4^-$ and N for $NH_3$.

but instead to the concentrations after some of the clusters have fragmented in the inlet of the mass spectrometer. In fact, the acid and base monomer concentration dependence is very similar for the measured concentrations of the three smallest ions, $HSO_4^- \cdot (H_2SO_4)_{0-2}$, which would be consistent with some of the trimers being detected as monomers and dimers after having fragmented inside the instrument.

## 4.2 Evaporation rates from the analysis

Figure 4 shows the posterior distributions of the coefficients corresponding to logarithms of the evaporation rates. The three sets of distributions correspond to different options for treating the background ammonia concentration. Either all below-



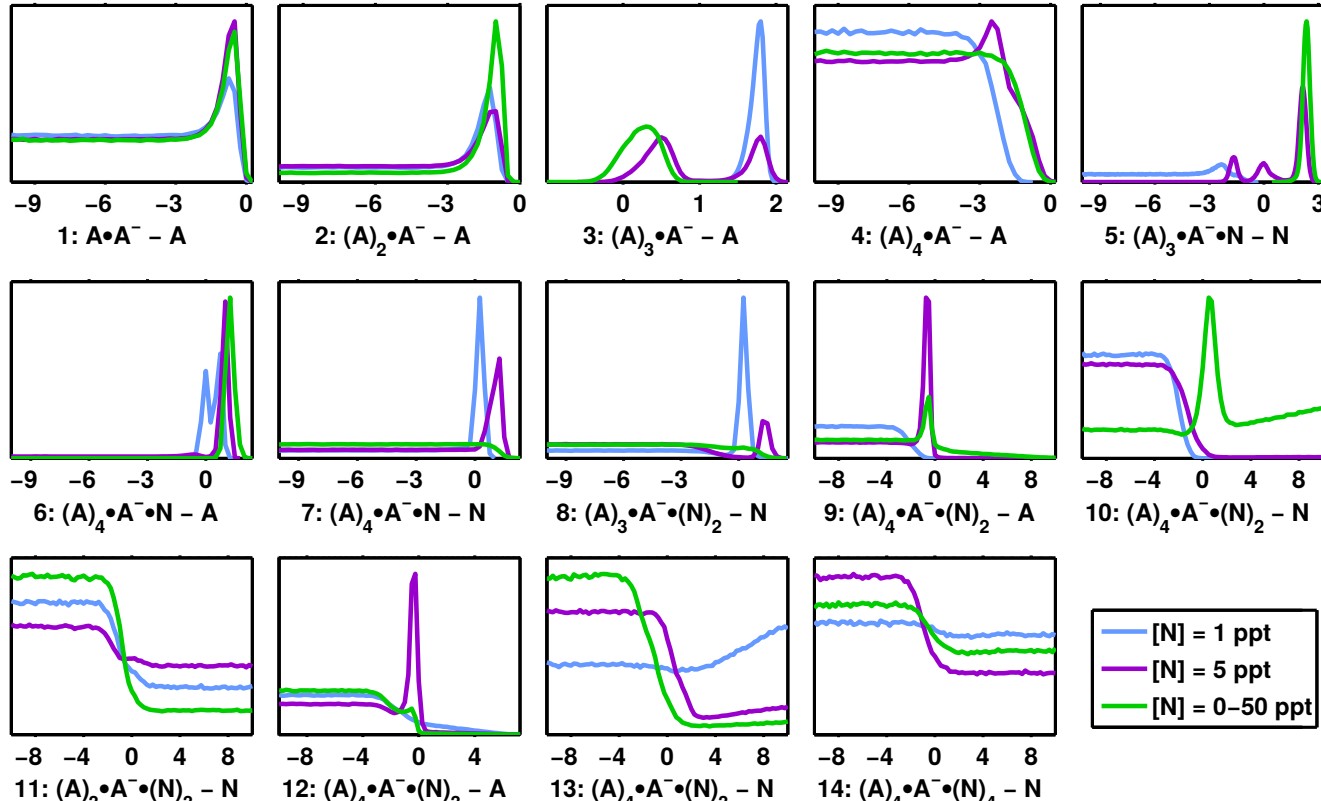

**Figure 4.** Posterior distributions of the base 10 logarithm of the evaporation rates (in units of $s^{-1}$) corresponding to the experimental cluster distributions and different options for treating the background ammonia concentration in the experiments where it was below the detection limit and therefore unknown. A stands for $H_2SO_4$, $A^-$ for $HSO_4^-$ and N for $NH_3$.

detection-limit ammonia concentrations are varied separately as MCMC parameters (green), or they are all set to 1 ppt (blue) or 5 ppt (purple). In the MCMC simulation where the background ammonia concentration is fitted, the median values for these concentrations are between 7 and 20, although the values are spread from 0 to 30 or 40.

All sets of MCMC simulations give a similar result for parameters number 1, 2 and 4: the pure negatively charged sulfuric
5   acid dimer $HSO_4^- \cdot H_2SO_4$, trimer $HSO_4^- \cdot (H_2SO_4)_2$ and pentamer $HSO_4^- \cdot (H_2SO_4)_4$ are stable, having evaporation rates below
$1\,s^{-1}$. The reason for the uniform shape of these distributions at low evaporation rates is that once the evaporation rate is much lower than the rates of any competing processes, its exact value has no effect on the cluster distribution. As discussed in the Supplementary Material, the peak seen in some of these distributions should not be interpreted as giving a good estimate for the evaporation rate – instead, the evaporation rate can have any value below the threshold where the probability density goes
10   to zero.



The distributions of some of the other evaporation rates depend strongly on the ammonia concentration assumed for the low-ammonia experiments. For instance, an ammonia concentration of 10 or 20 ppt (corresponding to the case where the ammonia concentrations were treated as free parameters) would require the $HSO_4^- \cdot (H_2SO_4)_3 \cdot NH_3$ cluster to have an ammonia evaporation rate of about 200 s$^{-1}$ in order for the enough pure sulfuric acid tetramers $HSO_4^- \cdot (H_2SO_4)_3$ to be observed, while

the evaporation rate would need to be well below 1 s$^{-1}$ if the ammonia concentration was instead 1 ppt. A similar pattern is observed for some of the other ammonia evaporation rates, and interdependencies between the different evaporation rates lead to the posterior distributions of some sulfuric acid evaporation rates also depending on how the background ammonia concentration is treated

For the two cases where the background ammonia concentration is set to a fixed value, some of the posterior distributions

consist of several peaks (see Fig. 4). As described in more detail in the Supplementary Material, the MCMC results can in fact be divided into two or three separate solutions, respectively, for the cases with background ammonia concentrations of 1 ppt and 5 ppt. These alternative solutions correspond to different cluster types being stable and unstable, but they all still give an equally good fit to the measured cluster distributions. Plots of the posterior distributions corresponding to the different solutions are presented in Sect. S3 of the Supplementary Material.

The estimates extracted for the evaporation rates from the MCMC simulations are presented in Table 1. As discussed above, only an upper limit can be determined for some evaporation rates, and it should be noted that the actual value could equally well be just below this limit or several orders of magnitude lower. For example, the rate at which $HSO_4^- \cdot H_2SO_4$ dimers are lost through collisions with neutral sulfuric acid molecules is between 0.04 and 1.3 s$^{-1}$ in the different experiments. If the evaporation rate of the dimer is lower than this, the dimers will practically never evaporate before colliding with an $H_2SO_4$.

If the evaporation process never happens, its rate cannot be expected to be determined based on the measurements. In order to constrain these low evaporation rates more tightly, experiments with very low but well quantified precursor concentrations would be needed, resulting in a lower rate for the competing growth process, but external losses and collisions with positive ions would still limit the range of evaporation rates that can be determined.

For certain evaporation rates, a distinct peak is observed in the posterior distribution. Also in this case it should be kept

in mind that the true value could be anywhere within the width of the peak. As can be expected, all these well constrained evaporation rates are in the intermediate range, mostly between 1 and 100, where growth by collisions does not completely overwhelm the evaporation process, but the cluster is not so unstable that it would never collide and grow further. These clusters probably correspond to rate limiting steps on the main formation pathway.

Some of the parameters have posterior distributions with a non-zero probability density over the whole range. Some of these

evaporation processes occur between clusters that are grouped together in the cluster distribution, and others are perhaps not on the main formation pathway. In any case, they do not have a strong impact on how well the modeled concentrations fit to the experimental data, and their values are therefore not constrained.

Also evaporation rates estimated from quantum chemical Gibbs free energies Ortega et al. (2014) are presented in Table 1 for comparison. The theoretical evaporation rates have an uncertainty of one or two orders of magnitude, as they depend

exponentially on the stepwise cluster formation energies, which have an uncertainty of 1–2 kcal mol$^{-1}$. For the three smallest





| [NH$_3$] in MCMC | 1 ppt | 1 ppt | 5 ppt | 5 ppt | 5 ppt | 0–50 ppt | | |
|---|---|---|---|---|---|---|---|---|
| | (A) | (B) | (C) | (D) | (E) | | QC |
| 1: A · A$^-$ − A | <1 | <0.6 | <1 | <1 | <1 | <1 | 8×10$^{-18}$ |
| 2: A$_2$ · A$^-$ − A | <0.2 | <0.2 | <0.2 | <0.2 | <0.3 | <0.3 | 2×10$^{-4}$ |
| 3: A$_3$ · A$^-$ − A | **60** | **60** | **60** | **60** | **3** | **2** | 1 |
| | (20–90) | (30–90) | (20–100) | (8–100) | (0.5–20) | (0.4–7) | |
| 4: A$_4$ · A$^-$ − A | <.01 | <0.03 | <0.03 | <0.06 | <0.6 | <0.3 | 200 |
| 5: A$_3$ · A$^-$ · N − N | <0.2 | <0.03 | **0.02** | **1** | **100** | **200** | 2 |
| | | | (<0.1) | (0.1–20) | (20–600) | (20–800) | |
| 6: A$_4$ · A$^-$ · N − A | **1** | **5** | **8** | <3 | **10** | **20** | 0.08 |
| | (<2) | (2–10) | (<20) | | (<30) | (3–100) | |
| 7: A$_4$ · A$^-$ · N − N | **2** | <3 | **6** | **20** | <20 | <60 | 6×10$^{-4}$ |
| | (<6) | | (1–30) | (6–60) | | | |
| 8: A$_3$ · A$^-$ · N$_2$ − N | <10 | **2** | **20** | <30 | <2 | <60 | 0.5 |
| | | (<100) | (<200) | | | | |
| 9: A$_4$ · A$^-$ · N$_2$ − A | <0.1 | <0.2 | – | <1 | <1 | – | 0.002 |
| 10: A$_4$ · A$^-$ · N$_2$ − N | <0.1 | <0.3 | – | <10 | <0.6 | – | 0.01 |
| 11: A$_3$ · A$^-$ · N$_3$ − N | – | – | – | – | – | – | 200 |
| 12: A$_4$ · A$^-$ · N$_3$ − A | < 10$^6$ | < 10$^6$ | <6×10$^5$ | <6×10$^5$ | <3 | <3×10$^4$ | 3×10$^{-9}$ |
| 13: A$_4$ · A$^-$ · N$_3$ − N | – | – | – | – | – | – | 3×10$^{-4}$ |
| 14: A$_4$ · A$^-$ · N$_4$ − N | – | – | – | – | – | – | 9×10$^8$ |

**Table 1.** Evaporation rates corresponding to the three MCMC simulations presented in Fig. 4. The cases where the background ammonia is set to a fixed value of 1 ppt or 5 ppt are divided into two and three alternative solutions, respectively, denoted as (A)-(E). (See the Supplementary Material for more details.) For parameters that have a posterior distribution with a clear peak and practically zero probability density elsewhere, the location of the peak is given together with the range of possible values in parentheses. In many cases only an upper limit can be determined, and some rates cannot be determined at all (–). The last column presents quantum-chemistry based evaporation rates for comparison. In the reactions, A stands for H$_2$SO$_4$, A$^-$ for HSO$_4^-$ and N for NH$_3$.

pure sulfuric acid clusters, HSO$_4^-$ · (H$_2$SO$_4$)$_{1-3}$, the quantum chemistry–based evaporation rates are in good agreement with the values determined from analyzing the experimental data. The pure acid pentamer HSO$_4^-$ · (H$_2$SO$_4$)$_4$, on the other hand, is predicted by quantum chemistry to have an evaporation lifetime of only 5 ms, while the analysis of the experimental data suggests that it has a very low evaporation rate (and hence a very long evaporation lifetime). In case of the ammonia-containing

5  clusters, the MCMC simulations with different options concerning the background ammonia concentration, as well as the different alternative solutions from the simulations, give different ranges of most likely values of the evaporation rates, some of which agree better and some worse with the theoretical estimates.



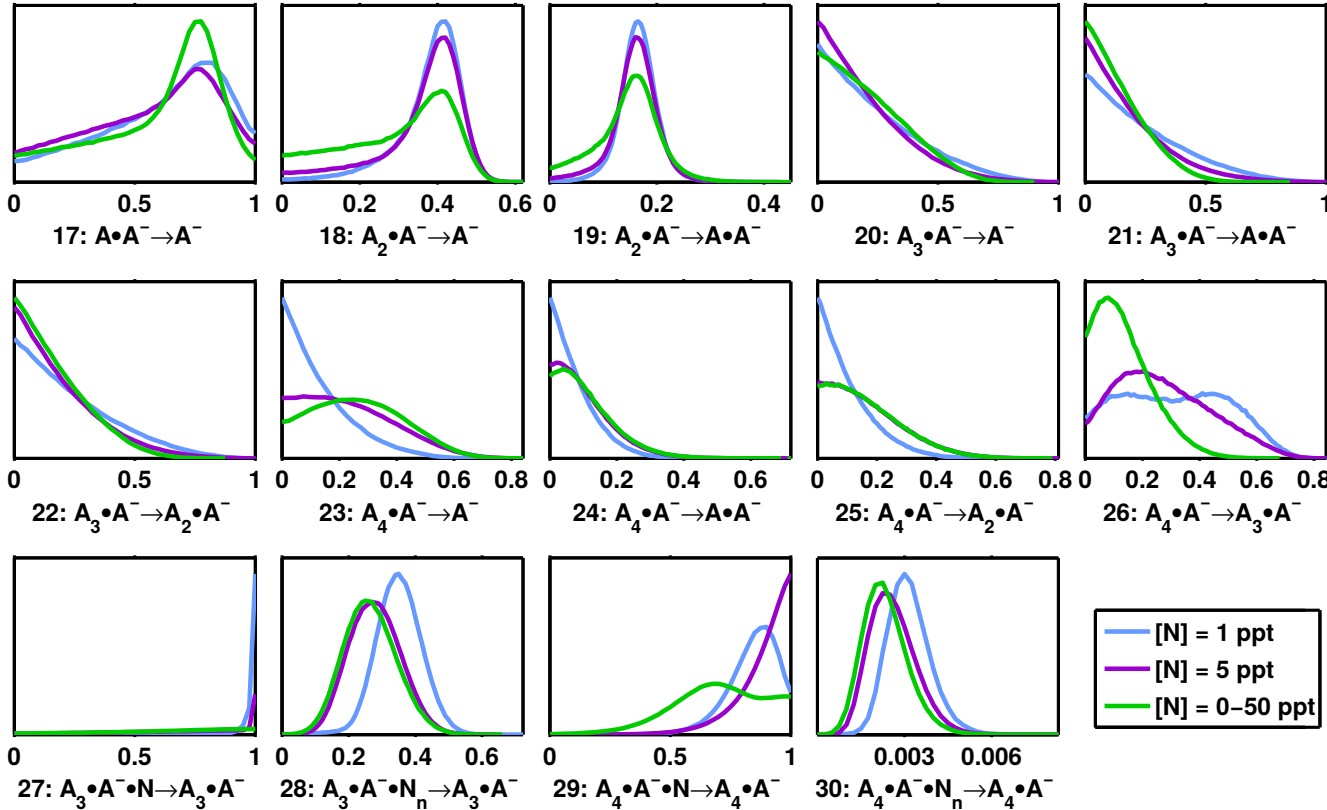

**Figure 5.** Posterior distributions of the fragmentation probabilities in the mass spectrometer inlet corresponding to the experimental cluster distributions and different options for treating the background ammonia concentration in the experiments where it was below the detection limit and therefore unknown. A stands for $H_2SO_4$, $A^-$ for $HSO_4^-$ and N for $NH_3$.

## 4.3 Estimating fragmentation probabilities

The probabilities of fragmentation processes that might occur in the inlet of the mass spectrometer were varied separately from the evaporation rates, as the process involved is different: the evaporation rates discussed in the previous section correspond to molecules evaporating spontaneously from the cluster at atmospheric pressure and a temperature of 273 K, while fragmenta-

5  tion in the inlet occurs when the ionic clusters are accelerated and experience high-energy collisions with neutral carrier gas molecules. In reality, the two concepts are not totally unrelated, as both processes depend on the binding energy of the cluster, but the fragmentation probability is also likely to depend on the number of vibrational degrees of freedom that can absorb energy from the collision. As the different factors determining the fragmentation probability, and even the exact conditions inside the APi-TOF inlet, remain unclear, all fragmentation probabilities were varied freely.

10  Figure 5 shows posterior distributions for the studied fragmentation probabilities. For the larger pure acid clusters $HSO_4^- \cdot$ $(H_2SO_4)_{2-4}$, several different fragmentation processes are considered, and their probabilities are presented separately in Fig.




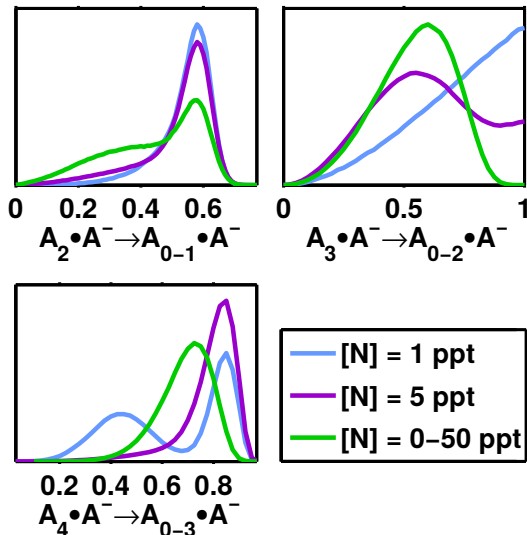

**Figure 6.** Posterior distributions of the total fragmentation probabilities of the $HSO_4^- \cdot (H_2SO_4)_{2\text{-}4}$ clusters corresponding to the experimental cluster distributions and different options for treating the background ammonia concentration in the experiments where it was below the detection limit and therefore unknown. A stands for $H_2SO_4$ and $A^-$ for $HSO_4^-$.

5. The overall fragmentation probabilities of these clusters are shown in Fig. 6. For the MCMC simulations with a fixed background ammonia concentration, the distributions corresponding to the alternative solutions (see previous Section) are shown in Sect. S3 of the Supplementary Material.

The posterior distribution of the dimer fragmentation probability is spread over the whole range from no fragmentation to 100% fragmentation. While there is a peak close to 70%, the possibility of dimers not fragmenting at all (which seems likely based on earlier experimental and theoretical evidence of the dimer being extremely stable) is not ruled out.

The trimers are found to fragment to some extent, producing both monomers and dimers. Assuming that both dimers and trimers have very low evaporation rates, but the tetramer is not very stable, the $HSO_4^-$ ions that are formed from charging $H_2SO_4$ molecules will quickly gain first one and then a second acid molecule and, as the next growth step is slower, accumulate to form a high concentration of trimers. If a notable fraction of these trimers fragments both into monomers and dimers, most of the monomers and dimers that are detected may actually be fragmentation products from trimers, as was also observed experimentally by Adamov et al. (2013). This would mean that the actual concentrations of negatively charged monomers and dimers in the chamber cannot be measured, preventing the accurate determination of the dimer evaporation rate and fragmentation probability. This scenario is in good agreement with the observations that the dimer fragmentation probability cannot be determined and only a relatively high upper limit can be found for the dimer evaporation rate.

Also the pure acid tetramers and pentamers fragment, possibly even more than the trimers, but it cannot be determined which fragmentation pathways are most important. A large fraction of the $HSO_4^- \cdot (H_2SO_4)_{3\text{-}4} \cdot NH_3$ clusters probably lose the ammonia molecule before detection, although the exact shape of the posterior distributions depends on how the low ammonia



concentrations are treated in the MCMC simulation. The results for the probability of the clusters containing two or more ammonia molecules losing all of them, on the other hand, is almost independent of the simulation options, and only a small fraction of these clusters are detected as pure acid clusters.

## 5 Conclusions

A Markov chain Monte Carlo (MCMC) approach is presented for determining evaporation rates from measured cluster distributions. The time evolution of the cluster population is described by birth-death equations and solved numerically. The values of the collision and evaporation rates are varied, and the obtained cluster distributions are compared to the measurements. In addition to the evaporation rates, also several other poorly known parameters related to the experimental setup are varied. The method is applied to concentration distributions of negatively charged sulfuric acid–ammonia clusters measured in the CLOUD chamber in CERN.

Of the pure sulfuric acid ion clusters $HSO_4^- \cdot (H_2SO_4)_{1\text{-}4}$, the dimer, trimer and pentamer are found to be very stable, while the tetramer has a higher evaporation rate and may correspond to a rate limiting step in the cluster formation process. The stability of the dimer and trimer and the instability of the tetramer are consistent with cluster formation energies calculated with different quantum chemical methods (Ortega et al., 2014; Herb et al., 2013) and with semi-empirical estimates combining measurements and quantum chemistry (Lovejoy and Curtius, 2001; Curtius et al., 2001). However, the low evaporation rate of the pure acid pentamer is in contradiction with the computational and semi-empirical cluster energies (Ortega et al., 2014; Lovejoy and Curtius, 2001; Curtius et al., 2001). On the other hand, these previously determined cluster energies correspond to dry clusters, and hydration is likely to stabilize clusters at least to some extent. Furthermore, evaporation rates calculated based on cluster formation energies involve the assumption that the evaporation process proceeds directly from the minimum energy configuration of the initial cluster to the minimum energy configuration of the product cluster. In reality, the process is likely to require some reorganization of the molecules and might have an energy barrier that slows down the evaporation.

The results are more ambiguous for the ammonia containing clusters. The MCMC simulations produce several alternative sets of evaporation rates that all provide an equally good fit to the experimental cluster distributions. This inconclusiveness stems at least partly from the choice of ammonia concentrations used in the set of experiments. In more than half of the experiments, the ammonia concentrations are in an unknown narrow range below the detection limit of 35 ppt, while the other runs have ammonia concentrations in a second narrow range from 100 to 250 ppt. Repeating the MCMC simulations with a new set of experimental cluster distributions measured at ammonia concentrations distributed evenly over a wide range would most probably narrow down the estimates for many of the evaporation rates.

The observation that several alternative sets of parameter values can produce a good fit to the same experimental data highlights the risk in using a simplified cluster model with only one or two fitting parameters as was done by Jen et al. (2014) and Kürten et al. (2015). While the model may give a good fit to the observations, the corresponding set of evaporation rates may be only one out of several solutions, and does not necessarily correspond to the true evaporation rates.





Another important finding is that fragmentation in the inlet of a APi-TOF mass spectrometer may have a significant effect on the observed cluster distribution. The amount of fragmentation depends on the type of inlet that is used, and also the specific voltages and other settings that are used. However, if it is not possible to suppress fragmentation completely for some instrument type or experimental setup, it is important at least to gain some understanding of the fragmentation processes, and

5   MCMC analysis appears to be a suitable tool for this.

While definitive values could not yet be obtained for all evaporation rates, the MCMC approach is shown to be a promising new tool for analyzing cluster concentration measurements. It can give valuable information about cluster evaporation processes that cannot be observed directly. However, enough experimental data measured over a wide range of all precursor concentrations are needed in order to draw clear conclusions. All details related to the experimental setup must be mimicked

10   as closely as possible in the simulations in order for the fitting parameters to have a clear physical meaning. Furthermore, as cluster formation is inherently a dynamical process, the MCMC analysis would be more efficient for datasets of cluster concentrations as a function of time, instead of the steady-state distributions used here. This would also enable the fitting of collision rate constants in addition to evaporation rates.

*Acknowledgements.* I would like to thank CSC – IT Center for Science Ltd for computer resources, the Vilho, Yrjö and Kalle Väisälä

15   Foundation and the European Research Council (project ERC-StG 257360-MOCAPAF) for funding, and Prof. Hanna Vehkamäki, Dr. Tinja Olenius and Prof. Heikki Haario for useful discussions and comments regarding the manuscript.





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
