# Peer review of "A Monte Carlo approach for determining cluster evaporation rates from concentration measurements"

_Atmospheric Chemistry and Physics, 2016_

## Referee Comment (RC1) · A. Nadykto (Referee) · 17 May 2016

The manuscript presents a series of MCMC simulations aimed at determining cluster evaporation rates from concentration measurements. The topic of the paper is interesting and important . The paper's well-written and easy to follow. After a thorough validation, the proposed approach could possibly be developed into a useful theoretical tool linking cluster concentrations and evaporation rates. However, I have to recommend major revisions because the number of issues to be addressed before the paper can be further considered for publication is quite large and some of them are serious..

Comments

[Figure]

I. Introduction is a way too self-referential, dedicated almost exclusively to own work and fails to acknowledge important contributions made by others. It also contains some misleading statements that need correction. 1.1 The clusters considered in the paper are relevant directly to the Ion -Mediated Nucleation (IMN), which is an important source of new particles in the Earth's atmosphere ( see e.g. Geophys. Res. Lett., 27, 883-886, 2000; J. Geophy. Res., 106, 4797-4814, 2001; Atmos. Chem. Phys., 8, 2537-2554, 2008; Atmos. Chem. Phys., 12, 11451-11463, 2012). A brief discussion on these matters accompanied by the corresponding references should be included in the Introduction to the revised manuscript. 1.2. The discussion on quantum-chemical studies on charged sulfuric acid-ammonia and sulfuric acid-ammonia-water clusters is limited to Almeida et al., 2013; Olenius et al., 2013b and fails to acknowledge a number of relevant contributions made by others (e.g. JPC A 116(24) 5886-5899, 2011; J. Phys. Chem., A, 117, 133-152, 2013; Atmos. Chem. Phys., 9, 4031-4038, 2009; PCCP, 10, 7073 - 7078, 2008). References to the aforementioned and other relevant studies should be included in the revised manuscript. 1.3. MC has been widely used in nucleation and cluster formation research since 2000s. In particular, a well-known MC-based DNT (Dynamic Nucleation Theory) has been developed by Kathmann and Garrett with co-workers at the PNNL ( e. g. PRL82(17):3484-3487, 1999. JPC B 105(47):11719-11728, 2001, J.Chem. Phys. 120(19):9133-914, 2004; . It would be useful to include a brief discussion on earlier applications of MC to nucleation and cluster formation in the revised manuscript. 1.4. The statement that "At the same time, modeling of particle formation has also advanced greatly in the past few years. For the first time theoretical predictions of cluster distributions (Olenius et al., 2013b) and particle formation rates (Almeida et al., 2013) agree qualitatively with experimental findings." is partly misleading because predictions of particle formation rates in Almeida et al., 2013 clearly disagree with the experimental data (Chem. Phys. Lett„ 624, 111-118, 2015). The statement should be corrected. 1.5. The author states that "This approach has been shown to give qualitative agreement with experiments (Almeida et al., 2013; Olenius et al., 2013b), but several very drastic assumptions are

involved. First-principles molecular dynamics simulations (Loukonen et al., 2014a, b) have shown that one harmonically oscillating cluster structure is far from a realistic description of the thermal motion of molecules in a cluster, implying that the traditional way of computing cluster formation free energies may be a rough approximation". However, this statement is obviously misleading because conclusions obtained using lower level theory such as ab initio MD Loukonen et al., 2014a, b are not applicable to results obtained using higher level theory such as ab initio or DFT. Unharmonic corrections for DFT level with typical scaling factors of 0.95-0.99 are very low and cannot significantly impact cluster formation rates. Also , the impacts of local minima on resulting thermochemical properties can be easily calculated using the Gibbs-Boltzmann distribution. This statement should be either modified or deleted. II. The source of thermochemical data used for computing evaporation rates in Table 1 is unclear. The MC fitting data were compared to Ortega et al. (2014) only. The author states that "Also evaporation rates estimated from quantum chemical Gibbs free energies Ortega et al. (2014) are presented in Table 1 for comparison". However, I wasn't able to find any data on Gibbs free energies in Ortega et al. (2014). Neither delta H nor delta S values were found in there. Delta G values seem to be missing in Ortega et al. (2014), too. Please, clarify the source of the data and include computations of evaporation rates based on quantum data obtained by others in Table 1 of your paper. III. Temperature dependency of evaporation rates is very important; however, the analysis of the temperature-dependent evaporation rates is missing. I would suggest the author to perform a study of evaporation rates for a few clusters at the room temperature and T=273.15 K and compare MC fitted evaporation rates with those obtained using quantum methods in Ortega et al. (2014) and other related studies. IV. It is well-known that uncertainties in measured cluster concentrations may be pretty big due to impurities, charging and other issues. The influence of the experimental uncertainties on MC fitted evaporation rates and fragmentation in the mass spectrometer should be discussed in some detail.

---

## Referee Comment (RC2) · Anonymous Referee #2 · 2 Jul 2016

Kupiainen-Määttä presents a Markov chain Monte Carlo (MCMC) study to derive sets of evaporation rates from observed cluster distributions of negatively charged sulfuric acid ammonia clusters. The simulations are expanded by also treating the fragmentation rates of the clusters in the mass spectrometer as unknown parameters that are varied with MCMC as well.

The paper is generally well written. It presents a useful modelling exercise to gain insight into cluster evaporation rates that are difficult to access. The MCMC is especially useful to realize that several different sets of fitting parameters are well suited to describe a set of experimental cluster measurements, and finding one well-fitting solution does not necessarily mean that this is the correct set of parameters. Exploring MCMC for this type of data is valuable. For larger data sets, covering larger ranges of conditions, hopefully in the future more and more firm conclusions can be drawn from this type of analysis.

The paper is publishable in ACP after addressing the following comments:

1) p1 l12: The Sipilä et al. 2010 paper is not a good reference for this statement because it claimed that the H2SO4/H2O system alone would be sufficient to explain the nucleation rates as observed in the BL.

2) p1 l17: The high res ToF mass spectrometers certainly allowed a lot of advances for characterizing the clusters during nucleation, but also earlier MS studies such as described by Hanson and Eisele, JGR, 2000 and 2002, already allowed to study the first steps of cluster formation for the sulfuric acid/water and sulfuric acid/ammonia systems.

3) p2 l26 and line 30/31: Besides Olenius et al., 2013b, also other references for the CLOUD data should be included: At least Kirkby et al., Nature, 2011, Schobesberger et al., ACP, 2015, and Duplissy et al., JGR, 2016, should be cited here as well. These papers are from the experimental groups and describe the experimental set-up and the experimental data in much more detail. Referencing only Olenius et al. does not give credit to the many other groups that contributed in order to set up and perform the CLOUD experiments and to obtain the experimental data that are used here (note, for example, that only authors from U Helsinki are part of Olenius et al. but many more groups were involved running the experiments and obtaining the H2SO4 and NH3 concentrations that are used here).

4) p5 l110: The assumption of a size-independent wall loss coefficient is problematic. The diffusion coefficient is strongly size dependent, and a cluster consisting of 5 sulfuric acid molecules will diffuse much slower to the walls than the monomer or dimer. This needs to be mentioned, and it should be discussed in how far it may influence the results.

5) Section 3.2.1. and Section 4.3: Besides fragmentation also the transmission efficiency of the mass spectrometer should be discussed (see, e.g. Heinritzi et al., AMT, 2016). The mass dependent transmission efficiency also influences the observed cluster distributions. While fragmentation can only lead to an overestimation of the measured small clusters and underestimation of the large clusters, changes in the transmission efficiency can also have the opposite effect. Transmission efficiency is very dependent on the tuning of the individual mass spectrometer. Influences on the observed distributions due to uncertainties of the transmission efficiency or mass discrimination should be discussed.

6) Section 3.3. At some point the limits of the MCMC should be discussed in more detail. Currently this discussion is distributed over the paper and limitations become evident from the results but it would be helpful to state the limitations already in the beginning of Section 3.3. When just 22 experimental distributions can be used to derive a large set of parameters, and additionally the input parameters are correlated, then the solutions will not be unambiguous. More discussion of this is needed.

7) Table 1 and section 4.2: The "alternative solutions" and cases (A)-(E) are listed but not explained at all. The differences need to be briefly described so that the reader has some idea about what is different in these cases without reading the Supplementary Material (see also comment #11).

8) Figure 6 shows the total fragmentation probabilities, e.g. the upper left panel, displaying $A_2A^-$ $\rightarrow A_{0-1}A^-$, should be formed from #18 and #19 from Fig 5. Why does the peak at about 0.2, where #19 has its maximum, not show up in the upper left panel of Fig 6? Adding a scale to the y-axis could be helpful.

9) P17 l11-21: a) An unexpected result is the high stability of the pentamer while the tetramer is less stable. It is mentioned that the stability could be due to hydration of the pentamer but hydration should also stabilize the tetramer. Please discuss.
   b) Could it be that the pentamer forms in a "closed shell" cluster configuration that is more stable than the tetramer?
   c) The stabilities can also compared with the lifetimes of clusters discussed in Hanson and Eisele, JGR, 2002, Section 2.3.2 and 3.1.

10) Acknowledgment: p18 l14-16. The CLOUD team and CERN resources should be acknowledged for provision of the experimental data.

11) Supplementary Material. I am lost in section S2.6. It is not clear how the separation was made and why it was made in the way it was made. At the end of p7 the separation of several cases is briefly explained. I do not understand why parameters 3 and 5 are selected for the separation of the synthetic data and why is parameter 6 selected for the posterior distributions with 1 ppt ammonia and parameter 5 for the 5 ppt ammonia simulations, respectively.
   It is stated that "First, it can be seen in Fig. S4 that the posterior distribution of coefficient number 3 has two peaks." I think, Figure 4 is meant here. But even then, only the purple line (5ppt) has two peaks (are we supposed to look only at the puple line? Why not blue and green?). The selection process seems to be arbitrary.
   Furthermore, the five lines of description of S3.3 on page 9 are much too short. It is still unclear what makes the difference for cases (A) and (B), and (C)-(E).

12) Section S2.6. Second line: "consider a case were" $\rightarrow$ "consider a case where"

---

## Author Comment (AC1) · 13 Aug 2016

I would like to thank Dr. Nadykto for his comments, which helped me improve the manuscript. Below are my point-by-point answers (normal font) to the comments (bold font) as well as the additions (yellow highlight) made to the manuscript (italic font). The line numbers refer to the revised manuscript.

**The manuscript presents a series of MCMC simulations aimed at determining cluster evaporation rates from concentration measurements. The topic of the paper is interesting and important . The paper's well-written and easy to follow. After a thorough validation, the proposed approach could possibly be developed into a useful theoretical tool linking cluster concentrations and evaporation rates. However, I have to recommend major revisions because the number of issues to be addressed before the paper can be further considered for publication is quite large and some of them are serious..**

Comments

**I. Introduction is a way too self-referential, dedicated almost exclusively to own work and fails to acknowledge important contributions made by others. It also contains some misleading statements that need correction.**

There were 15 references in the Introduction, and only two of them were papers where I am a co-author. I have trouble seeing this as "way too self-referential". The new count after the revisions I have made is 19 references including 4 where I am a co-author, which still seems quite reasonable.

**1.1 The clusters considered in the paper are relevant directly to the Ion -Mediated Nucleation (IMN), which is an important source of new particles in the Earth's atmosphere ( see e.g. Geophys. Res. Lett., 27, 883-886, 2000; J. Geophy. Res., 106, 4797-4814, 2001; Atmos. Chem. Phys., 8, 2537-2554, 2008; Atmos. Chem. Phys., 12, 11451-11463, 2012). A brief discussion on these matters accompanied by the corresponding references should be included in the Introduction to the revised manuscript.**

A mention of ions and ionic clusters was indeed missing. However, as Yu and Turco (2000) were neither the first to suggest ion-induced cluster formation nor the first to demonstrate it experimentally, I decided to cite the CLOUD experiments instead.

Page 1, lines 15–18:
"*The experiments of Kirkby et al. (2011); Almeida et al. (2013) have also shown that the first steps of cluster formation can proceed along an ionic pathway, and that this process can dominate over the electrically neutral pathway when there are not enough base molecules or other impurities available to stabilize the small neutral sulfuric acid clusters.*"

**1.2. The discussion on quantum-chemical studies on charged sulfuric acid-ammonia and sulfuric acid-ammonia-water clusters is limited to Almeida et al., 2013; Olenius et al., 2013b and fails to acknowledge a number of relevant contributions made by others (e.g. JPC A 116(24) 5886-5899, 2011; J. Phys. Chem., A, 117, 133-152, 2013; Atmos. Chem. Phys., 9, 4031-4038, 2009; PCCP, 10, 7073 - 7078, 2008). References to the aforementioned and other relevant studies should be included in the revised manuscript.**

The discussion about which is the best quantum chemistry method for atmospherical clusters has been going on more than long enough (Nadykto et al., Entropy 2011, 13, 554–569; Kurtén, Entropy 2011, 13, 915–923; Nadykto et al., Nadykto et al., Chem. Phys. Lett. 2014, 609, 42–49; Kupiainen-Määttä et al., Chem. Phys. Lett. 2015, 624, 107–110), and I see no reason to continue it. As the cluster energies cannot be measured directly, there is no way to find out which method gives the

best predictions, or whether there even is any method that could be trusted. The whole point of this paper is to find a new way to obtain information on cluster properties, so that we no longer need to rely on quantum chemistry calculations at all. I have now tried to explain this more clearly in the Introduction.

Page 2, lines 11–15:
"*As evaporation rates depend exponentially on the cluster formation energies, theoretical evaporation rates may easily be wrong by several orders of magnitude. ==Different quantum chemistry methods can give qualitatively very different predictions for cluster concentrations (Kupiainen-Määttä et al., 2013; Kupiainen-Määttä et al., 2015), and it is not clear whether any of the methods can be trusted.== Also the treatment of the collision rates is highly simplified, but errors of more than a factor of two or perhaps ten are unlikely.*"

The quantum chemistry data is used only to provide a test case for the MCMC data analysis method. I could just as well have used some other quantum chemistry data set or simply random numbers, but it seemed more sensible to use cluster energies that reproduce the measured cluster distributions qualitatively, if not quantitatively. However, the test data is not claimed to mimic perfectly the true cluster concentrations.

**1.3. MC has been widely used in nucleation and cluster formation research since 2000s. In particular, a well-known MC-based DNT (Dynamic Nucleation Theory) has been developed by Kathmann and Garrett with co-workers at the PNNL ( e. g. PRL82(17):3484-3487, 1999. JPC B 105(47):11719-11728, 2001, J.Chem. Phys. 120(19):9133-914, 2004; . It would be useful to include a brief discussion on earlier applications of MC to nucleation and cluster formation in the revised manuscript.**

In DNT, Monte Carlo has been used for computing an integral. This is different from using Monte Carlo for parameter estimation as is done in the present paper.

**1.4. The statement that "At the same time, modeling of particle formation has also advanced greatly in the past few years. For the first time theoretical predictions of cluster distributions (Olenius et al., 2013b) and particle formation rates (Almeida et al., 2013) agree qualitatively with experimental findings." is partly misleading because predictions of particle formation rates in Almeida et al., 2013 clearly disagree with the experimental data (Chem. Phys. Lett,, 624, 111-118, 2015). The statement should be corrected.**

The results are stated to agree qualitatively. This does not imply that they would match perfectly.

**1.5. The author states that "This approach has been shown to give qualitative agreement with experiments (Almeida et al., 2013; Olenius et al., 2013b), but several very drastic assumptions are involved. First-principles molecular dynamics simulations (Loukonen et al., 2014a, b) have shown that one harmonically oscillating cluster structure is far from a realistic description of the thermal motion of molecules in a cluster, implying that the traditional way of computing cluster formation free energies may be a rough approximation". However, this statement is obviously misleading because conclusions obtained using lower level theory such as ab initio MD Loukonen et al., 2014a, b are not applicable to results obtained using higher level theory such as ab initio or DFT. Unharmonic corrections for DFT level with typical scaling factors of 0.95-0.99 are very low and cannot significantly impact cluster formation rates. Also , the impacts of local minima on resulting thermochemical properties can be easily calculated using the Gibbs-Boltzmann distribution. This statement should be either modified or deleted.**

The simulations of Loukonen et al. used DFT to compute the energies and forces, so it is unclear

why the conclusions "are not applicable to results obtained using higher level theory such as [...] DFT**".** In any case, the problem is not so much the anharmonicity of vibrations within or even between the molecules, but the observation that the molecules rotate inside the cluster breaking bonds and forming new ones. This has been clarified in the revised paper.

Page 2, lines 8–11:
" *First-principles molecular dynamics simulations (Loukonen et al., 2014a, b) have shown that one harmonically oscillating cluster structure is far from a realistic description of the thermal motion of molecules in a cluster, ==as molecules may rotate inside the cluster continuously breaking intermolecular bonds and forming new ones. This implies== that the traditional way of computing cluster formation free energies may be a rough approximation.*"

**II. The source of thermochemical data used for computing evaporation rates in Table 1 is unclear. The MC fitting data were compared to Ortega et al. (2014) only. The author states that "Also evaporation rates estimated from quantum chemical Gibbs free energies Ortega et al. (2014) are presented in Table 1 for comparison". However, I wasn't able to find any data on Gibbs free energies in Ortega et al. (2014). Neither delta H nor delta S values were found in there. Delta G values seem to be missing in Ortega et al. (2014), too. Please, clarify the source of the data and include computations of evaporation rates based on quantum data obtained by others in Table 1 of your paper.**

The evaporation rates were discussed by Ortega et al., while the cluster energies were published already by Almeida et al. (2013). This has been corrected in the revised manuscript.

Page 15, lines 15–16:
"*Also evaporation rates estimated from quantum chemical Gibbs free energies (Ortega et al., 2014==; Almeida et al., 2013==) are presented in Table 1 for comparison.*"

Unfortunately there don't seem to exist any other data sets with quantum chemical Gibbs free energies for all the relevant clusters, which could be included in the Table. Herb et al. (2013) have done calculations on some of the smaller clusters, and these results were used as a comparison already in the submitted version of the manuscript.

Page 18, lines 6–9:
"*The stability of the dimer and trimer and the instability of the tetramer are consistent with cluster formation energies calculated with different quantum chemical methods (Ortega et al., 2014; Herb et al., 2013) and with semi-empirical estimates combining measurements and quantum chemistry (Lovejoy and Curtius, 2001; Curtius et al., 2001).*"

**III. Temperature dependency of evaporation rates is very important; however, the analysis of the temperaturedependent evaporation rates is missing. I would suggest the author to perform a study of evaporation rates for a few clusters at the room temperature and T=273.15 K and compare MC fitted evaporation rates with those obtained using quantum methods in Ortega et al. (2014) and other related studies.**

Studying the temperature dependence of the evaporation rates would  certainly be interesting, but it cannot be done using MCMC analysis without experimental input data. Olenius et al. (2013) do present some data also at 292 K and 248 K, but the data sets are unfortunately too small.

**IV. It is well-known that uncertainties in measured cluster concentrations may be pretty big due to impurities, charging and other issues. The influence of the experimental uncertainties on MC fitted evaporation rates and fragmentation in the mass spectrometer should be**

**discussed in some detail.**

It is unclear how impurities and charging would cause uncertainties in these measurements: the clusters are detected with a high-resolution mass spectrometer, so any impurities in them could not go unnoticed, and the clusters are ionic to begin with, so they do not need to be charged before detection.

---

## Author Comment (AC2) · 13 Aug 2016

I would like to thank Referee #2 for his/her thorough reading of the paper and very useful and constructive comments. Below are my point-by-point answers (normal font) to the comments (bold font) as well as the additions (yellow highlight) made to the manuscript (italic font). The line numbers refer to the revised manuscript.

**Kupiainen-Määttä presents a Markov chain Monte Carlo (MCMC) study to derive sets of evaporation rates from observed cluster distributions of negatively charged sulfuric acid ammonia clusters. The simulations are expanded by also treating the fragmentation rates of the clusters in the mass spectrometer as unknown parameters that are varied with MCMC as well.**

**The paper is generally well written. It presents a useful modelling exercise to gain insight into cluster evaporation rates that are difficult to access. The MCMC is especially useful to realize that several different sets of fitting parameters are well suited to describe a set of experimental cluster measurements, and finding one well-fitting solution does not necessarily mean that this is the correct set of parameters. Exploring MCMC for this type of data is valuable. For larger data sets, covering larger ranges of conditions, hopefully in the future more and more firm conclusions can be drawn from this type of analysis. The paper is publishable in ACP after addressing the following comments:**

**1) p1 l12: The Sipilä et al. 2010 paper is not a good reference for this statement because it claimed that the H2SO4/H2O system alone would be sufficient to explain the nucleation rates as observed in the BL.**

I would expect that undetected impurities in the laboratory air affected the particle formation rate, but perhaps it is indeed better to leave this reference out. The sentence was modified accordingly.

Page 1, lines 11–15:
“*Recent laboratory experiments (Berndt et al., 2010; Sipilä et al., 2010; Benson et al., 2011; Almeida et al., 2013) have confirmed that particle formation rates of the magnitude observed in the atmosphere can be produced with ambient sulfuric acid concentrations and either impurities present in laboratory air or intentionally added low concentrations of base molecules, giving new support for sulfuric acid being at least one of the compounds driving atmospheric particle formation.*”

**2) p1 l17: The high res ToF mass spectrometers certainly allowed a lot of advances for characterizing the clusters during nucleation, but also earlier MS studies such as described by Hanson and Eisele, JGR, 2000 and 2002, already allowed to study the first steps of cluster formation for the sulfuric acid/water and sulfuric acid/ammonia systems.**

The sentence has been modified and new references have been added.

Page 1, lines 19–21:
“*The development of highly sensitive mass spectrometers has enabled the detection and characterization of individual ionic clusters consisting of only a few molecules (Eisele and Hanson, 2000; Zhao et al., 2010; Junninen et al., 2010), opening a new window into the first steps of cluster formation.*”

**3) p2 l26 and line 30/31: Besides Olenius et al., 2013b, also other references for the CLOUD data should be included: At least Kirkby et al., Nature, 2011, Schobesberger et al., ACP, 2015, and Duplissy et al., JGR, 2016, should be cited here as well. These papers are from the experimental groups and describe the experimental set-up and the experimental data in much**

**more detail. Referencing only Olenius et al. does not give credit to the many other groups that contributed in order to set up and perform the CLOUD experiments and to obtain the experimental data that are used here (note, for example, that only authors from U Helsinki are part of Olenius et al. but many more groups were involved running the experiments and obtaining the H2SO4 and NH3 concentrations that are used here).**

A reference to Kirkby et al. (2011) was added, as that paper describes the experiments from which the data used of Olenius et al. was obtained. The later CLOUD publications are, to my knowledge, related to other later campaigns.

Page 2, lines 32–34:
"*The method is applied to measurement data from the CLOUD experiment (Olenius et al., 2013b; see Kirkby et al., 2011 for more details on the CLOUD experiment).*"

**4) p5 l110: The assumption of a size-independent wall loss coefficient is problematic. The diffusion coefficient is strongly size dependent, and a cluster consisting of 5 sulfuric acid molecules will diffuse much slower to the walls than the monomer or dimer. This needs to be mentioned, and it should be discussed in how far it may influence the results.**

The size dependence was already mentioned, but I have now explained in more detail that this seemed to be the least bad option.

Page 5, lines 16–22:
"*A wall loss rate of $1.7 \times 10^{-3}$ $s^{-1}$ was determined for the electrically neutral $H_2SO_4$ monomer in the CLOUD chamber (Almeida et al., 2013). This rate decreases with increasing cluster size, but ions may have a higher loss rate. The probability of an individual cluster being lost on a wall also varies with location inside the chamber, or in practice with time as the air is continuously circulated around the chamber by large fans. As the size, charge and composition dependence of the wall losses is not known, all clusters were, for simplicity, assumed to have the same wall loss rate, and its value was sampled from the range 0 and $10^{-2}$ $s^{-1}$. The size-independence of the wall loss rate may cause some uncertainty to the results, but introducing even more free parameters in order to vary the value separately for each cluster would also be problematic.*"

**5) Section 3.2.1. and Section 4.3: Besides fragmentation also the transmission efficiency of the mass spectrometer should be discussed (see, e.g. Heinritzi et al., AMT, 2016). The mass dependent transmission efficiency also influences the observed cluster distributions. While fragmentation can only lead to an overestimation of the measured small clusters and underestimation of the large clusters, changes in the transmission efficiency can also have the opposite effect. Transmission efficiency is very dependent on the tuning of the individual mass spectrometer. Influences on the observed distributions due to uncertainties of the transmission efficiency or mass discrimination should be discussed.**

This is a good point and should certainly be taken into account in future studies. In this paper, however, adding new fitting parameters is really not possible due to the small number of experimental data points. The transmission efficiency is now mentioned in the revised manuscript.

Page 18, line 32 – page 19, line 2:
"*However, if it is not possible to suppress fragmentation completely for some instrument type or experimental setup, it is important at least to gain some understanding of the fragmentation processes, and MCMC analysis appears to be a suitable tool for this. In this study, the mass spectrometer was assumed to have been calibrated so that there was no mass discrimination, but in the future, also the mass dependent transmission efficiency of mass spectrometers could be studied*"

*using MCMC analysis.*"

**6) Section 3.3. At some point the limits of the MCMC should be discussed in more detail. Currently this discussion is distributed over the paper and limitations become evident from the results but it would be helpful to state the limitations already in the beginning of Section 3.3. When just 22 experimental distributions can be used to derive a large set of parameters, and additionally the input parameters are correlated, then the solutions will not be unambiguous. More discussion of this is needed.**

This is not really a limitation of MCMC, but a problem caused by not having enough experimental data available. Therefore the problem is mentioned in Sect. 4 instead of Sect. 3.3 as suggested.

Page 9, lines 28–32:
"*Even when the MCMC simulation finds a good fit to the observed distributions, the interpretation of the output parameter distributions is not always clear. The number of input data points from the CLOUD experiment is so small that unambiguous values were not reached for most of the evaporation rates. To get better insight into what conclusions can safely be drawn, Sect. S2 of the Supplementary Material presents test simulations for synthetic input cluster distributions with known evaporation rates and fragmentation probabilities.*"

**7) Table 1 and section 4.2: The "alternative solutions" and cases (A)-(E) are listed but not explained at all. The differences need to be briefly described so that the reader has some idea about what is different in these cases without reading the Supplementary Material (see also comment #11).**

An explanation has been added.

Page 13, line 1 – page 14, line 3
"*These alternative solutions correspond to different cluster types being stable and unstable, but they all still give an equally good fit to the measured cluster distributions. For instance, when assuming a background ammonia concentration of 5 ppt, the posterior distribution of parameter number 5 shows three separate peaks (purple line in Fig. 4). Looking only at the sets of parameter values in the right hand side peak, it can be noted that the value of parameter number 3 corresponds always to the left hand side peak of this distribution (see Fig. S14 of the Supplementary Material), and parameter number 8 always has a low value due to correlations between the different parameters. This set of ranges for the parameter values is denoted as solution (E), and similarly the two other peaks in the distribution of parameter number 5 correspond to solutions (C) and (D). The observation that the distributions can be divided into separate solutions in this way implies that, for instance, either an evaporation rate of 100 $s^{-1}$ for ammonia from the $HSO_4^- \cdot (H_2SO_4)_3 \cdot NH_3$ cluster and an evaporation rate of 3 $s^{-1}$ of the pure sulfuric acid tetramer or an evaporation rate of 0.2 $s^{-1}$ for ammonia from the $HSO_4^- \cdot (H_2SO_4)_3 \cdot NH_3$ cluster and an evaporation rate of 60 $s^{-1}$ of the pure sulfuric acid tetramer could produce a good fit to the experimental cluster distributions, but an evaporation rate of 100 $s^{-1}$ for ammonia from the $HSO_4^- \cdot (H_2SO_4)_3 \cdot NH_3$ cluster and an evaporation rate of 60 $s^{-1}$ of the pure sulfuric acid tetramer would not reproduce the data. For the simulations with an ammonia concentration of 1 ppt, the separate solutions (A) and (B) correspond to the two peaks in the distribution of coefficient number 6.*"

**8) Figure 6 shows the total fragmentation probabilities, e.g. the upper left panel, displaying $A_2A^- \rightarrow A_{0-1}A^-$, should be formed from #18 and #19 from Fig 5. Why does the peak at about 0.2, where #19 has its maximum, not show up in the upper left panel of Fig 6? Adding a scale to the y-axis could be helpful.**

It should be kept in mind that the distribution in Fig. 6 is not the sum of the *distributions* in Fig. 5 but a distribution of the *sum of the parameters*. This has now been written out more clearly. The scale of the *y*-axis depends only on the number of iterations in the MCMC simulation, and is completely irrelevant.

Page 16, lines 3–4:
"*The posterior distributions of the overall fragmentation probabilities (that is the sums of the probabilities of all fragmentation processes in which a given cluster can be lost) of these clusters are shown in Fig. 6.*"

**9) P17 l11-21:**
**a) An unexpected result is the high stability of the pentamer while the tetramer is less stable. It is mentioned that the stability could be due to hydration of the pentamer but hydration should also stabilize the tetramer. Please discuss.**

What I meant is that perhaps the pentamer is stabilized more strongly by hydration than the other clusters. This has now been clarified in the text.

Page 18, lines 11–13:
"*On the other hand, these previously determined cluster energies correspond to dry clusters, and hydration is likely to stabilize clusters at least to some extent. It is, in principle, possible that the pentamer could have a very stable hydrated structure, while the tetramer would only be moderately stabilized by hydration.*"

**b) Could it be that the pentamer forms in a "closed shell" cluster configuration that is more stable than the tetramer?**

This is possible, although it would be somewhat surprising as it has not been observed in quantum chemistry studies. However, this would still not explain the discrepancy with the semi-empirical results of Lovejoy and Curtius (2001) and Curtius et al. (2001).

**c) The stabilities can also compared with the lifetimes of clusters discussed in Hanson and Eisele, JGR, 2002, Section 2.3.2 and 3.1.**

The reason why I had not discussed these results is that Hanson and Eisele only obtained estimates for the lifetimes of the clusters $HSO_4^-\cdot(H_2SO_4)_3\cdot(NH3)_3$ and $HSO_4^-\cdot(H_2SO_4)_4\cdot(NH3)_4$, while no estimates were obtained for the evaporation rates of these clusters in the present study. Thus there is nothing to compare.

**10) Acknowledgment: p18 l14-16. The CLOUD team and CERN resources should be acknowledged for provision of the experimental data.**

This is quite a strange suggestion, as I have used previously published data and cited the article from which I took that data, and I have had no contact whatsoever with the CLOUD community about this study.

**11) Supplementary Material. I am lost in section S2.6. It is not clear how the separation was made and why it was made in the way it was made. At the end of p7 the separation of several cases is briefly explained. I do not understand why parameters 3 and 5 are selected for the separation of the synthetic data and why is parameter 6 selected for the posterior distributions with 1 ppt ammonia and parameter 5 for the 5 ppt ammonia simulations, respectively. It is stated that "First, it can be seen in Fig. S4 that the posterior distribution of**

**coefficient number 3 has two peaks.” I think, Figure 4 is meant here. But even then, only the purple line (5ppt) has two peaks (are we supposed to look only at the puple line? Why not blue and green?). The selection process seems to be arbitrary. Furthermore, the five lines of description of S3.3 on page 9 are much too short. It is still unclear what makes the difference for cases (A) and (B), and (C)-(E).**

Some clarifications were added.

Page 7, 2nd paragraph:
“*The correlations can be found by grouping the sets of parameter values according to the value of one specific parameter, and comparing the posterior distributions of the other parameters for these groups. ==If the groups of parameters form completely separate peaks also in the distributions of one or more of the other parameters, a correlation has been found.==*”

Page 7, last paragraph:
“*First, it can be seen in Fig. S4 that the posterior distribution of coefficient number 3 has two peaks ==(light green line in Fig. S4).==*”

Page 9, last paragraph – page 10:
“*Figures S13-S18 show the posterior distributions of Figs. 4, 5 and 6 of the main article corresponding to MCMC simulations with a fixed background ammonia concentration separated into the alternative solutions (A)–(E) of Table 1. The posterior distributions were split into the different solutions similarly as described in Sect. S2.6. ==For the simulation with 1 ppt, the solutions were separated based on the two peaks in the distribution of coefficient number 6 (blue line in Fig. 4 of the main text), as this produced a neat separation also for coefficients number 7 and 8 (Fig. S13). For the case with 5 ppt ammonia the solutions were separated based on the three peaks in the distribution of coefficient number 5 (purple line in Fig. 4 of the main text), as this produced neat separation also for coefficients number 3, 6, 7 and 8 (Fig. S14).==*”

**12) Section S2.6. Second line: “consider a case were” → “consider a case where”**

The typo was corrected.

---

## Author Comment (AC4) · 13 Aug 2016

[revised manuscript text omitted]

Figure S8 demonstrates the separation of the MCMC results into three scenarios for the synthetic cluster concentration data and a fixed background ammonia concentration of 5 ppt in the MCMC simulation. For clarity, the distributions of the scenarios are normalized to have the same overall probability.

First, it can be seen in Fig. S4 that the posterior distribution of coefficient number 3 has two peaks (light green line in Fig. S4). The points in the smaller peak, that is the sets of parameter

[revised manuscript text omitted]

---

## Referee Report (RR1)

Reviewer 1 (Stage 2)

**COMMENTS**

The new Reviewer's comments (Italic), Author's answers (normal font) to the original comments (bold font), Author's additions (yellow highlight) made at Stage 1 and  the reference list are given below.

**The manuscript presents a series of MCMC simulations aimed at determining cluster evaporation rates from concentration measurements. The topic of the paper is interesting and important . The paper's well-written and easy to follow. After a thorough validation, the proposed approach could possibly be developed into a useful theoretical tool linking cluster concentrations and evaporation rates. However, I have to recommend major revisions because the number of issues to be addressed before the paper can be further considered for publication is quite large and some of them are serious.. Comments**

**I. Introduction is a way too self-referential, dedicated almost exclusively to own work and fails to acknowledge important contributions made by others. It also contains some misleading statements that need correction.**

There were 15 references in the Introduction, and only two of them were papers where I am a coauthor. I have trouble seeing this as "way too self-referential". The new count after the revisions I have made is 19 references including 4 where I am a co-author, which still seems quite reasonable.

*While 18 of 35 studies cited in the manuscript (over 50%) are the ones produced the Helsinki group to which the Author belongs, the Author is still not willing to acknowledge relevant contributions by others (see the Author's response to Comment 1.1 and others). This approach is far from the scholarly one and shall be corrected prior to publication.*

**1.1 The clusters considered in the paper are relevant directly to the Ion -Mediated Nucleation (IMN), which is an important source of new particles in the Earth's atmosphere ( see e.g. Geophys. Res. Lett., 27, 883-886, 2000; J. Geophy. Res., 106, 4797-4814, 2001; Atmos. Chem. Phys., 8, 2537-2554, 2008; Atmos. Chem. Phys., 12, 11451-11463, 2012). A brief discussion on these matters accompanied by the corresponding references should be included in the Introduction to the revised manuscript. A mention of ions and ionic clusters was indeed missing. However, as Yu and Turco (2000) were neither the first to suggest ion-induced cluster formation nor the first to demonstrate it experimentally, I decided to cite the CLOUD experiments instead.**

Page 1, lines 15–18: "The experiments of Kirkby et al. (2011); Almeida et al. (2013) have also shown that the first steps of cluster formation can proceed along an ionic pathway, and that this process can dominate over the electrically neutral pathway when there are not enough base molecules or other impurities available to stabilize the small neutral sulfuric acid clusters."

*It is important to note that Yu and Turco (2000) were first to demonstrate the relevance of ions to atmospheric nucleation and to show that IMN, which involves not only ions but also neutrals, is an important source of secondary aerosols in the Earth's atmosphere. In fact, they have shown that "the first steps of cluster formation can proceed along an ionic pathway" and that "that this process can dominate over the electrically neutral pathway when there are not enough base molecules or other impurities" over a decade earlier than Kirkby et al. (2011) and Almeida et al. (2013).   Their original*

*work and other papers on IMN suggested by the Reviewer are well-known in the field and relevant directly to the manuscript being reviewed, and, thus, they shall be properly cited and briefly discussed in the revised manuscript.*

**1.2. The discussion on quantum-chemical studies on charged sulfuric acid-ammonia and sulfuric acid-ammonia-water clusters is limited to Almeida et al., 2013; Olenius et al., 2013b and fails to acknowledge a number of relevant contributions made by others (e.g. JPC A 116(24) 5886-5899, 2011; J. Phys. Chem., A, 117, 133-152, 2013; Atmos. Chem. Phys., 9, 4031- 4038, 2009; PCCP, 10, 7073 - 7078, 2008). References to the aforementioned and other relevant studies should be included in the revised manuscript.**

The discussion about which is the best quantum chemistry method for atmospherical clusters has been going on more than long enough (Nadykto et al., Entropy 2011, 13, 554–569; Kurtén, Entropy 2011, 13, 915–923; Nadykto et al., Nadykto et al., Chem. Phys. Lett. 2014, 609, 42–49; Kupiainen-Määttä et al., Chem. Phys. Lett. 2015, 624, 107–110), and I see no reason to continue it. As the cluster energies cannot be measured directly, there is no way to find out which method gives the best predictions, or whether there even is any method that could be trusted. The whole point of this paper is to find a new way to obtain information on cluster properties, so that we no longer need to rely on quantum chemistry calculations at all. I have now tried to explain this more clearly in the Introduction.

Page 2, lines 11–15: "As evaporation rates depend exponentially on the cluster formation energies, theoretical evaporation rates may easily be wrong by several orders of magnitude. Different quantum chemistry methods can give qualitatively very different predictions for cluster concentrations (Kupiainen-Määttä et al., 2013; Kupiainen-Määttä et al., 2015), and it is not clear whether any of the methods can be trusted. Also the treatment of the collision rates is highly simplified, but errors of more than a factor of two or perhaps ten are unlikely."

The quantum chemistry data is used only to provide a test case for the MCMC data analysis method. I could just as well have used some other quantum chemistry data set or simply random numbers, but it seemed more sensible to use cluster energies that reproduce the measured cluster distributions qualitatively, if not quantitatively. However, the test data is not claimed to mimic perfectly the true cluster concentrations.

"As evaporation rates depend exponentially on the cluster formation energies, theoretical evaporation rates may easily be wrong by several orders of magnitude. Different quantum chemistry methods can give qualitatively very different predictions for cluster concentrations (Kupiainen-Määttä et al., 2013; Kupiainen-Määttä et al., 2015), and it is not clear whether any of the methods can be trusted. Also the treatment of the collision rates is highly simplified, but errors of more than a factor of two or perhaps ten are unlikely."

*Several claims made in the response to Comment 1.2: "…cluster energies cannot be measured directly.." , "there is no way to find out which method gives the best predictions… or … could be trusted", are obviously unjustified. First of all, the cluster energies have been being measured since 1960s and the fully referenced NIST Chemistry WebBook (NIST Standard Reference Database Number 69)* [http://webbook.nist.gov/chemistry/](http://webbook.nist.gov/chemistry/) *contain information on measured energies (enthalpies, entropies and Gibbs free energies) for several thousands of reactions involving ions that can be and are commonly used, alongside with higher-level ab initio studies, in validating DFT methods commonly used to study atmospheric clusters (see, for example, e.g. refs. [1-9] attached below). The relevant literature contains tons of benchmarking studies that are commonly used to justify the use of a specific*

*DFT method. Thirdly, the Author claims that errors in collision cross sections produced by kinetic models can be in error by a factor "of two or perhaps ten" . Where the "perhaps ten" is coming from?*

*Fourthly, the Author claims that "Different quantum chemistry methods can give qualitatively very different predictions for cluster concentrations" and cites two own papers (Kupiainen-Määttä et al., 2013; Kupiainen-Määttä et al., 2015 [10]) to support this claim. It is important to note that Kupiainen-Määttä et al., 2015 [10] is actually a Comment to Nadykto et al. 2014 [9], in which it was shown that anomalously large difference between the conventional quantum-chemical ab initio and DFT methods on one side, and the composite B3RICC2 method on the other side , is caused by the deficiency of the B3RICC2 method [11] developed and used by the Helsinki group ( see both Nadykto et al. 2014 [9], and Reply of Nadykto et al. (2015) [12] to Kupiainen-Määttä et al., 2015 [10], in which the conclusion about the deficiency of the B3RICC2 method has been fully confirmed).*

*This indicates that the paragraph*

*"As evaporation rates depend exponentially on the cluster formation energies, theoretical evaporation rates may easily be wrong by several orders of magnitude. Different quantum chemistry methods can give qualitatively very different predictions for cluster concentrations (Kupiainen-Määttä et al., 2013; Kupiainen-Määttä et al., 2015), and it is not clear whether any of the methods can be trusted. Also the treatment of the collision rates is highly simplified, but errors of more than a factor of two or perhaps ten are unlikely."*

*shall be either deleted or adequately revised, with acknowledging not only the Comment by Kupiainen-Määttä et al., 2015 [10] but also the original paper Nadykto et al. 2014 [9] and Reply to the Comment Nadykto et al. 2015 [11].*

**1.3. MC has been widely used in nucleation and cluster formation research since 2000s. In particular, a well-known MC-based DNT (Dynamic Nucleation Theory) has been developed by Kathmann and Garrett with co-workers at the PNNL ( e. g. PRL82(17):3484-3487, 1999. JPC B 105(47):11719-11728, 2001, J.Chem. Phys. 120(19):9133-914, 2004; . It would be useful to include a brief discussion on earlier applications of MC to nucleation and cluster formation in the revised manuscript.**

In DNT, Monte Carlo has been used for computing an integral. This is different from using Monte Carlo for parameter estimation as is done in the present paper.

*In my opinion, the difference in the way how MC is used does not justify not acknowledging the earlier relevant work. I think that Author shall include a brief discussion on the earlier MC studies in nucleation research and references to the aforementioned papers in the revised manuscript.*

**1.4. The statement that "At the same time, modeling of particle formation has also advanced greatly in the past few years. For the first time theoretical predictions of cluster distributions (Olenius et al., 2013b) and particle formation rates (Almeida et al., 2013) agree qualitatively with experimental findings." is partly misleading because predictions of particle formation rates in Almeida et al., 2013 clearly disagree with the experimental data (Chem. Phys. Lett,, 624, 111-118, 2015). The statement should be corrected.**

The results are stated to agree qualitatively. This does not imply that they would match perfectly.

*The claim that "particle formation rates (Almeida et al., 2013) agree qualitatively with experimental findings"  is strictly wrong because it has been shown (Nadykto et al (2015) [12] ) that particle formation rates computed based on erroneous (Nadykto et al (2014) [10], 2015 [12]) B3RICC2*

*thermochemistry [11] not only disagree with experimental nucleation rates data but also exhibit totally wrong, nearly zeros, dependency on amine concentrations [12]. Also, the "qualitative agreement" of "theoretical predictions of cluster distributions with experiments pointed out in the study of Olenius et al., 2013b, which is based on the very same erroneous B3RICC2 [11] thermochemistry, is a questionable achievement because the agreement may indicate problems in other parts of the computational methodology used in Olenius et al., 2013b that could led to the "qualitative agreement" due to the cancellation of errors only.*

*In the view of these circumstances, the discussion on "agreement" of Olenius et al., 2013b and Almeida et al., 2013 with experimental data shall be either properly revised or, preferably, deleted.*

1.5. **The author states that "This approach has been shown to give qualitative agreement with experiments (Almeida et al., 2013; Olenius et al., 2013b), but several very drastic assumptions are involved. First-principles molecular dynamics simulations (Loukonen et al., 2014a, b) have shown that one harmonically oscillating cluster structure is far from a realistic description of the thermal motion of molecules in a cluster, implying that the traditional way of computing cluster formation free energies may be a rough approximation". However, this statement is obviously misleading because conclusions obtained using lower level theory such as ab initio MD Loukonen et al., 2014a, b are not applicable to results obtained using higher level theory such as ab initio or DFT. Unharmonic corrections for DFT level with typical scaling factors of 0.95-0.99 are very low and cannot significantly impact cluster formation rates. Also , the impacts of local minima on resulting thermochemical properties can be easily calculated using the Gibbs-Boltzmann distribution. This statement should be either modified or deleted.**

The simulations of Loukonen et al. used DFT to compute the energies and forces, so it is unclear why the conclusions "are not applicable to results obtained using higher level theory such as [...] DFT". In any case, the problem is not so much the anharmonicity of vibrations within or even between the molecules, but the observation that the molecules rotate inside the cluster breaking bonds and forming new ones. This has been clarified in the revised paper.

Page 2, lines 8–11: ==「First-principles molecular dynamics simulations (Loukonen et al., 2014a, b) have shown that one harmonically oscillating cluster structure is far from a realistic description of the thermal motion of molecules in a cluster, as molecules may rotate inside the cluster continuously breaking intermolecular bonds and forming new ones. This implies that the traditional way of computing cluster formation free energies may be a rough approximation."==

*First of all, first-principles molecular dynamics simulations involve far large list of assumptions and approximations than conventional DFT, on which they are partly based. In addition to "first principles"/DFT, ab initio molecular dynamics involves lower-level classical theory ( "molecular dynamics simulations, where the atomic nuclei evolve in time according to the classical equations of motion", "GTH pseudo-potentials were used for the core electrons", simulation box was 20 Å X 20 Å X 20 Å , simulation times and many others ( see Simulations and Collision Simulations sections in Loukonen et al., 2014a, b [13,14]). Also, the computations were done for a single density functional, with no sensitivity studies of model results to the density fucntionals used, input parameters, basis sets, pseudopotentials, box size, cut-offs etc. carried out. Secondly, the computations in Loukonen et al., 2014a, b [13-14] represent for a limited set for small NEUTRAL CLUSTERS ONLY THAT ARE NOT RELEVANT TO THE IONIC CLUSTERS STUDIED HERE.*

*This shows clearly that the aforementioned papers of Loukonen et al., 2014a, b [13-14]   are inconclusive and irrelevant to the present study. This also indicates that attempted attacks on quantum-chemical methods  in the present work are unfounded and, thus, the statement*

==*"First-principles molecular dynamics simulations (Loukonen et al., 2014a, b) have shown that one harmonically oscillating cluster structure is far from a realistic description of the thermal motion of molecules in a cluster, as molecules may rotate inside the cluster continuously breaking intermolecular bonds and forming new ones. This implies that the traditional way of computing cluster formation free energies may be a rough approximation."*==

*shall be deleted.*

**II. The source of thermochemical data used for computing evaporation rates in Table 1 is unclear. The MC fitting data were compared to Ortega et al. (2014) only. The author states that "Also evaporation rates estimated from quantum chemical Gibbs free energies Ortega et al. (2014) are presented in Table 1 for comparison". However, I wasn't able to find any data on Gibbs free energies in Ortega et al. (2014). Neither delta H nor delta S values were found in there. Delta G values seem to be missing in Ortega et al. (2014), too. Please, clarify the source of the data and include computations of evaporation rates based on quantum data obtained by others in Table 1 of your paper.**

The evaporation rates were discussed by Ortega et al., while the cluster energies were published already by Almeida et al. (2013). This has been corrected in the revised manuscript.

==Page 15, lines 15–16: "Also evaporation rates estimated from quantum chemical Gibbs free energies (Ortega et al., 2014; Almeida et al., 2013) are presented in Table 1 for comparison." Unfortunately there don't seem to exist any other data sets with quantum chemical Gibbs free energies for all the relevant clusters, which could be included in the Table. Herb et al. (2013) have done calculations on some of the smaller clusters, and these results were used as a comparison already in the submitted version of the manuscript.==

Page 18, lines 6–9: =="The stability of the dimer and trimer and the instability of the tetramer are consistent with cluster formation energies calculated with different quantum chemical methods (Ortega et al., 2014; Herb et al., 2013) and with semi-empirical estimates combining measurements and quantum chemistry (Lovejoy and Curtius, 2001; Curtius et al., 2001)."==

**III. Temperature dependency of evaporation rates is very important; however, the analysis of the temperature dependent evaporation rates is missing. I would suggest the author to perform a study of evaporation rates for a few clusters at the room temperature and T=273.15 K and compare MC fitted evaporation rates with those obtained using quantum methods in Ortega et al. (2014) and other related studies.**

Studying the temperature dependence of the evaporation rates would certainly be interesting, but it cannot be done using MCMC analysis without experimental input data. Olenius et al. (2013) do present some data also at 292 K and 248 K, but the data sets are unfortunately too small.

**IV. It is well-known that uncertainties in measured cluster concentrations may be pretty big due to impurities, charging and other issues. The influence of the experimental uncertainties on MC fitted evaporation rates and fragmentation in the mass spectrometer should be discussed in some detail.**

It is unclear how impurities and charging would cause uncertainties in these measurements: the clusters are detected with a high-resolution mass spectrometer, so any impurities in them could not go unnoticed, and the clusters are ionic to begin with, so they do not need to be charged before detection.

*Actually, there exist a number of sources of large uncertainties in measured particle number concentrations. Some of them have already been pointed out by the Author in the Experimental Cluster Distribution Section (page 3):*

"The clusters were detected using a high resolution APi-TOF (Atmospheric Pressure interface Time-Of-Flight) mass spectrometer. The largest clusters considered in the study contained one $HSO_4$ – ion, four $H_2SO_4$ molecules and four ammonia molecules. However, it is likely that most of the clusters initially also contained some water molecules, although none were detected, and water was concluded to evaporate from the clusters inside the APi-TOF. The clusters were also assumed to lose some or all of the ammonia molecules inside the instrument prior to detection."

*where uncertainties related to critically important hydration effect and lost of ammonia prior to detection are clearly acknowledged. These uncertainties may have a very large impact on measured particle formation rates, and, thus, some estimates of their impacts shall be included in the revised manuscript. The Author could use factors of 10 and 100 as the model "typical uncertainties" in measured particle number concentrations.*

*References*

1. *Husar, D. E., Temelso, B., Ashworth, A. L., & Shields, G. C. (2012). Hydration of the bisulfate ion: atmospheric implications. The Journal of Physical Chemistry A, 116(21), 5151-5163.*
2. *Temelso, Berhane, Thuong Ngoc Phan, and George C. Shields. "Computational study of the hydration of sulfuric acid dimers: Implications for acid dissociation and aerosol formation." The Journal of Physical Chemistry A 116.39 (2012): 9745-9758.*
3. *Pickard, F. C., Griffith, D. R., Ferrara, S. J., Liptak, M. D., Kirschner, K. N., & Shields, G. C. (2006). CCSD (T), W1, and other model chemistry predictions for gas-phase deprotonation reactions. International journal of quantum chemistry, 106(15), 3122-3128.*
4. *Temelso, B., Archer, K. A., & Shields, G. C. (2011). Benchmark structures and binding energies of small water clusters with anharmonicity corrections.The Journal of Physical Chemistry A, 115(43), 12034-12046.*
5. *Bork, N., Du, L., Reiman, H., Kurtén, T., & Kjaergaard, H. G. (2014). Benchmarking ab initio binding energies of hydrogen-bonded molecular clusters based on FTIR spectroscopy. The Journal of Physical Chemistry A,118(28), 5316-5322.*
6. *Elm, J., Bilde, M., & Mikkelsen, K. V. (2013). Assessment of binding energies of atmospherically relevant clusters. Physical Chemistry Chemical Physics, 15(39), 16442-16445.*
7. *Anacker, Tony, and Joachim Friedrich. "New accurate benchmark energies for large water clusters: DFT is better than expected." Journal of computational chemistry 35, 8 (2014), 634-643.*
8. *Sherrill, C. D., Takatani, T., & Hohenstein, E. G. (2009). An Assessment of Theoretical Methods for Nonbonded Interactions: Comparison to Complete Basis Set Limit Coupled-Cluster Potential Energy Curves for the Benzene Dimer, the Methane Dimer, Benzene– Methane, and Benzene– H2S†. The Journal of Physical Chemistry A, 113(38), 10146-10159*
9. *Nadykto, A. B., Herb, J., Yu, F., & Xu, Y. (2014). Enhancement in the production of nucleating clusters due to dimethylamine and large uncertainties in the thermochemistry of amine-enhanced nucleation.Chemical Physics Letters, 609, 42-49.*
10. *Kupiainen-Määttä, Oona, et al. "Comment on 'Enhancement in the production of nucleating clusters due to dimethylamine and large uncertainties in the thermochemistry of amine-enhanced nucleation'by Nadykto et al., Chem. Phys. Lett. 609 (2014) 42–49." Chemical Physics Letters 624 (2015): 107-110.*
11. *I.K. Ortega, et al., Atmos. Chem. Phys. 12 (2012) 225–235, http://dx.doi.org/10.5194/acp-12-225-2012*

12. Nadykto, A. B., Herb, J., Yu, F., Nazarenko, E. S., & Xu, Y. (2015). Reply to the 'Comment on "Enhancement in the production of nucleating clusters due to dimethylamine and large uncertainties in the thermochemistry of amine-enhanced nucleation"'by Kupiainen-Maatta et al. Chemical Physics Letters,624, 111-118.
13. Loukonen, V., Bork, N., and Vehkamäki, H.: From Collisions to Clusters: First Steps of Sulfuric Acid Nanocluster Formation Dynamics, Mol. Phys., 112, 1979 – 1986, doi:10.1080/00268976.2013.877167, 2014a.
14. Loukonen, V., Kuo, I.-F. W., McGrath, M. J., and Vehkamäki, H.: On the stability and dynamics of (sulfuric acid) (ammonia) and (sulfuric acid) (dimethylamine) clusters: A first-principles molecular dynamics investigation, Chem. Phys., 428, 164 – 174, doi:10.1016/j.chemphys.2013.11.014, 2014b.

---

## Author Response (AR2)

Dear Editor,

Below in green are my point-by point answers to Dr. Nadykto's new comments (italic font). Changes to the manuscript are highlighted in yellow. Some of the first round comments and responses have been left out, while those relevant for the new comments are presented in black bold font (referee comments) and black normal font (my earlier responses).

**I. Introduction is a way too self-referential, dedicated almost exclusively to own work and fails to acknowledge important contributions made by others. It also contains some misleading statements that need correction.**

There were 15 references in the Introduction, and only two of them were papers where I am a coauthor. I have trouble seeing this as "way too self-referential". The new count after the revisions I have made is 19 references including 4 where I am a co-author, which still seems quite reasonable.

*While 18 of 35 studies cited in the manuscript (over 50%) are the ones produced the Helsinki group to which the Author belongs, the Author is still not willing to acknowledge relevant contributions by others (see the Author's response to Comment 1.1 and others). This approach is far from the scholarly one and shall be corrected prior to publication.*

Some new references have been added. See responses to other comments below.

**1.1 The clusters considered in the paper are relevant directly to the Ion -Mediated Nucleation (IMN), which is an important source of new particles in the Earth's atmosphere ( see e.g. Geophys. Res. Lett., 27, 883-886, 2000; J. Geophy. Res., 106, 4797-4814, 2001; Atmos. Chem. Phys., 8, 2537-2554, 2008; Atmos. Chem. Phys., 12, 11451-11463, 2012). A brief discussion on these matters accompanied by the corresponding references should be included in the Introduction to the revised manuscript.**

A mention of ions and ionic clusters was indeed missing. However, as Yu and Turco (2000) were neither the first to suggest ion-induced cluster formation nor the first to demonstrate it experimentally, I decided to cite the CLOUD experiments instead.

Page 1, lines 15–18:
"The experiments of Kirkby et al. (2011); Almeida et al. (2013) have also shown that the first steps of cluster formation can proceed along an ionic pathway, and that this process can dominate over the electrically neutral pathway when there are not enough base molecules or other impurities available to stabilize the small neutral sulfuric acid clusters."

*It is important to note that Yu and Turco (2000) were first to demonstrate the relevance of ions to atmospheric nucleation and to show that IMN, which involves not only ions but also neutrals, is an important source of secondary aerosols in the Earth's atmosphere. In fact, they have shown that "the first steps of cluster formation can proceed along an ionic pathway" and that "that this process can dominate over the electrically neutral pathway when there are not enough base molecules or other impurities" over a decade earlier than Kirkby et al. (2011) and Almeida et al. (2013). Their original work and other papers on IMN suggested by the Reviewer are well-known in the field and relevant directly to the manuscript being reviewed, and, thus, they shall be properly cited and briefly discussed in the revised manuscript.*

A reference was added to Yu and Turco (2000).

Page 1, lines 15-20:

"==Also ions have been suggested to play a role in atmospheric cluster formation (Yu and Turco, 2000), as ions are produced constantly by cosmic rays and radon decay, and small ionic clusters are more stable than their neutral counterparts.== The experiments of Kirkby et al. (2011) and Almeida et al. (2013) have ==recently== shown that the first steps of cluster formation can ==indeed== proceed along an ionic pathway, and that this process can dominate over the electrically neutral pathway when there are not enough base molecules or other impurities available to stabilize the small neutral sulfuric acid clusters."

**[...]**

*Several claims made in the response to Comment 1.2: "...cluster energies cannot be measured directly..", "there is no way to find out which method gives the best predictions... or ... could be trusted", are obviously unjustified. First of all, the cluster energies have been being measured since 1960s*

Cluster energies cannot be measured. What is measured are cluster concentrations. Converting these into cluster energies is nontrivial as discussed in the present manuscript.

In cases where clusters are very unstable and do not grow to large sizes even at high vapor concentrations, the system is close to equilibrium and cluster energies can be solved directly from the cluster concentrations. Such systems were indeed studied already in the 1960s, but even then the assumptions required in the analysis and the resulting uncertainties were acknowledged and discussed in detail (see eg. Hogg et al. 1966, J. Am. Chem. Soc., 88(1):28 – 31) but this discussion seems unfortunately to have been forgotten since.

For the clusters studied in the present manuscript, the situation is completely different. Solving cluster energies directly from cluster concentrations requires that the concentrations correspond to a dynamic equilibrium. Obtaining a dynamic equilibrium is simply not possible in the case of negatively charged sulfuric acid–ammonia clusters: at high vapor concentrations the clusters grow indefinitely and a constant source of vapor molecules is required to compensate for the formation of large particles, and on the other hand at low precursor concentrations losses to chamber walls cannot be ignored and once again a constant source of vapor molecules is needed to compensate for the losses. Neither of these systems is therefore a closed system, and thus cannot be in equilibrium. Therefore the approach used in the 1960s to obtain cluster energies cannot be used for sulfuric acid–ammonia clusters.

*and the fully referenced NIST Chemistry WebBook (NIST Standard Reference Database Number 69) http://webbook.nist.gov/chemistry/ contain information on measured energies (enthalpies, entropies and Gibbs free energies) for several thousands of reactions involving ions that can be and are commonly used, alongside with higher-level ab initio studies, in validating DFT methods commonly used to study atmospheric clusters (see, for example, e.g. refs. [1-9] attached below). The relevant literature contains tons of benchmarking studies that are commonly used to justify the use of a specific DFT method. Thirdly, the Author claims that errors in collision cross sections produced by kinetic models can be in error by a factor "of two or perhaps ten". Where the "perhaps ten" is coming from?*

This was simply a worst-case scenario, as the collision rates have not been studied very thoroughly. It has now been left out. (Page 2, line 19.)

*Fourthly, the Author claims that "Different quantum chemistry methods can give qualitatively very different predictions for cluster concentrations" and cites two own papers (Kupiainen-Määttä et*

*al., 2013; Kupiainen-Määttä et al., 2015 [10]) to support this claim.*

The reason for only citing my own papers is that **no one else has ever studied the effect of uncertainties in cluster energies on actual measurable cluster concentrations**. The constrained equilibrium concentrations shown in Fig. 4 of Nadykto et al. (2014) have nothing to do with actual measurable atmospheric cluster concentrations as we explain in our comment to that paper.

*It is important to note that Kupiainen-Määttä et al., 2015 [10] is actually a Comment to Nadykto et al. 2014 [9], in which it was shown that anomalously large difference between the conventional quantum-chemical ab initio and DFT methods on one side, and the composite B3RICC2 method on the other side , is caused by the deficiency of the B3RICC2 method [11] developed and used by the Helsinki group ( see both Nadykto et al. 2014 [9], and Reply of Nadykto et al. (2015) [12] to Kupiainen-Määttä et al., 2015 [10], in which the conclusion about the deficiency of the B3RICC2 method has been fully confirmed).*

Nadykto et al. (2014) indeed pointed out (once again) the differences between the **cluster energies** obtained using two different methods. This is well-known, and not the point of the paragraph of the present manuscript being discussed. They also only pointed out the difference in the energies, but did not have any convincing proof (such as comparisons with experiments) for their claim that their own energies were correct and ours were wrong. On the contrary, the PW91 functional used by Nadykto et al. fails to predict the observed high concentration of neutral sulfuric acid– dimethylamine clusters. A reference to an earlier paper presenting a more thorough comparison of different quantum chemistry methods has been added to the discussion.

Page 2, lines 14-17:
"As ==estimates of cluster formation energies based on different quantum chemical approaches may differ by several kcal/mol (Leverentz et al., 2013) and== evaporation rates depend exponentially on the cluster formation energies, theoretical evaporation rates may easily be wrong by several orders of magnitude."

*This indicates that the paragraph "As evaporation rates depend exponentially on the cluster formation energies, theoretical evaporation rates may easily be wrong by several orders of magnitude. Different quantum chemistry methods can give qualitatively very different predictions for cluster concentrations (Kupiainen-Määttä et al., 2013; Kupiainen-Määttä et al., 2015), and it is not clear whether any of the methods can be trusted. Also the treatment of the collision rates is highly simplified, but errors of more than a factor of two or perhaps ten are unlikely."shall be either deleted or adequately revised, with acknowledging not only the Comment by Kupiainen-Määttä et al., 2015 [10] but also the original paper Nadykto et al. 2014 [9] and Reply to the Comment Nadykto et al. 2015 [11].*

**1.3. MC has been widely used in nucleation and cluster formation research since 2000s. In particular, a well-known MC-based DNT (Dynamic Nucleation Theory) has been developed by Kathmann and Garrett with co-workers at the PNNL ( e. g. PRL82(17):3484-3487, 1999. JPC B 105(47):11719-11728, 2001, J.Chem. Phys. 120(19):9133-914, 2004; . It would be useful to include a brief discussion on earlier applications of MC to nucleation and cluster formation in the revised manuscript.**

In DNT, Monte Carlo has been used for computing an integral. This is different from using Monte Carlo for parameter estimation as is done in the present paper.

*In my opinion, the difference in the way how MC is used does not justify not acknowledging the earlier relevant work. I think that Author shall include a brief discussion on the earlier MC studies*

*in nucleation research and references to the aforementioned papers in the revised manuscript.*

I agree with the reviewer that "earlier **relevant** work" should be acknowledged. What he suggests is, however, acknowledging work that is **not relevant** for the present study. Similarly every time anyone publishes a paper where they optimize cluster structures, they would need to refer to all papers ever published where someone uses some optimization algorithm in any context whatsoever.

**1.4. The statement that "At the same time, modeling of particle formation has also advanced greatly in the past few years. For the first time theoretical predictions of cluster distributions (Olenius et al., 2013b) and particle formation rates (Almeida et al., 2013) agree qualitatively with experimental findings." is partly misleading because predictions of particle formation rates in Almeida et al., 2013 clearly disagree with the experimental data (Chem. Phys. Lett,, 624, 111-118, 2015). The statement should be corrected.**

The results are stated to agree qualitatively. This does not imply that they would match perfectly.

*The claim that "particle formation rates (Almeida et al., 2013) agree qualitatively with experimental findings" is strictly wrong because it has been shown (Nadykto et al (2015) [12] ) that particle formation rates computed based on erroneous (Nadykto et al (2014) [10], 2015 [12]) B3RICCthermochemistry [11] not only disagree with experimental nucleation rates data but also exhibit totally wrong, nearly zeros, dependency on amine concentrations [12]. Also, the "qualitative agreement" of "theoretical predictions of cluster distributions with experiments pointed out in the study of Olenius et al., 2013b, which is based on the very same erroneous B3RICC2 [11] thermochemistry, is a questionable achievement because the agreement may indicate problems in other parts of the computational methodology used in Olenius et al., 2013b that could led to the "qualitative agreement" due to the cancellation of errors only.*

*In the view of these circumstances, the discussion on "agreement" of Olenius et al., 2013b and Almeida et al., 2013 with experimental data shall be either properly revised or, preferably, deleted.*

The results were stated to agree qualitatively. This does not imply that they would match perfectly. This has been further emphasized.

Page 2, lines 1-4:
"At the same time, modeling of particle formation has also advanced greatly in the past few years. For the first time, simulations involving no empirical fitting parameters give qualitatively correct predictions for the sulfuric acid concentration dependence of cluster concentrations (Olenius et al., 2013b) and particle formation rates (Almeida et al., 2013), although quantitative agreement with experimental findings is still far from perfect."

**1.5. The author states that "This approach has been shown to give qualitative agreement with experiments (Almeida et al., 2013; Olenius et al., 2013b), but several very drastic assumptions are involved. First-principles molecular dynamics simulations (Loukonen et al., 2014a, b) have shown that one harmonically oscillating cluster structure is far from a realistic description of the thermal motion of molecules in a cluster, implying that the traditional way of computing cluster formation free energies may be a rough approximation". However, this statement is obviously misleading because conclusions obtained using lower level theory such as ab initio MD Loukonen et al., 2014a, b are not applicable to results obtained using higher level theory such as ab initio or DFT. Unharmonic corrections for DFT level with typical scaling factors of 0.95-0.99 are very low and cannot significantly impact cluster formation rates. Also , the impacts of local minima on resulting thermochemical properties can be easily calculated using the Gibbs-Boltzmann distribution.**

**This statement should be either modified or deleted.**

The simulations of Loukonen et al. used DFT to compute the energies and forces, so it is unclear why the conclusions "are not applicable to results obtained using higher level theory such as [...] DFT". In any case, the problem is not so much the anharmonicity of vibrations within or even between the molecules, but the observation that the molecules rotate inside the cluster breaking bonds and forming new ones. This has been clarified in the revised paper.

Page 2, lines 8–11: "First-principles molecular dynamics simulations (Loukonen et al., 2014a, b) have shown that one harmonically oscillating cluster structure is far from a realistic description of the thermal motion of molecules in a cluster, as molecules may rotate inside the cluster continuously breaking intermolecular bonds and forming new ones. This implies that the traditional way of computing cluster formation free energies may be a rough approximation."

*First of all, first-principles molecular dynamics simulations involve far large list of assumptions and approximations than conventional DFT, on which they are partly based. In addition to "first principles"/DFT, ab initio molecular dynamics involves lower-level classical theory ( "molecular dynamics simulations, where the atomic nuclei evolve in time according to the classical equations of motion", "GTH pseudo-potentials were used for the core electrons", simulation box was 20 Å X 20 Å X 20 Å , simulation times and many others ( see Simulations and Collision Simulations sections in Loukonen et al., 2014a, b [13,14]). Also, the computations were done for a single density functional, with no sensitivity studies of model results to the density fucntionals used, input parameters, basis sets, pseudopotentials, box size, cut-offs etc. carried out. Secondly, the computations in Loukonen et al., 2014a, b [13-14] represent for a limited set for small NEUTRAL CLUSTERS ONLY THAT ARE NOT RELEVANT TO THE IONIC CLUSTERS STUDIED HERE.*

*This shows clearly that the aforementioned papers of Loukonen et al., 2014a, b [13-14] are inconclusive and irrelevant to the present study. This also indicates that attempted attacks on quantum-chemical methods in the present work are unfounded and, thus, the statement*

*"First-principles molecular dynamics simulations (Loukonen et al., 2014a, b) have shown that one harmonically oscillating cluster structure is far from a realistic description of the thermal motion of molecules in a cluster, as molecules may rotate inside the cluster continuously breaking intermolecular bonds and forming new ones. This implies that the traditional way of computing cluster formation free energies may be a rough approximation."*

*shall be deleted.*

I fully agree with the reviewer that results obtained for neutral clusters are never directly applicable to ionic clusters. However, I do not agree that it would be better to ignore these results completely rather than be aware of them and consider what implications they might have. I also don't see any reason to believe that molecules would only be able to rotate in electrically neutral clusters and not in ionic ones. The text has been modified to clarify these points.

Page 2, lines 9-13:
"First-principles molecular dynamics simulations (Loukonen et al., 2014a, b) have shown that one harmonically oscillating cluster structure is far from a realistic description of the thermal motion of molecules in a ==small electrically neutral== cluster, as molecules may rotate inside the cluster continuously breaking intermolecular bonds and forming new ones. ==Although only electrically neutral clusters were studied, some of the sulfuric acid–dimethylamine clusters are very strongly bound, and similar processes might, therefore, take place also in strongly-bound ionic clusters.=="

**[…]**

**IV. It is well-known that uncertainties in measured cluster concentrations may be pretty big due to impurities, charging and other issues. The influence of the experimental uncertainties on MC fitted evaporation rates and fragmentation in the mass spectrometer should be discussed in some detail.**

It is unclear how impurities and charging would cause uncertainties in these measurements: the clusters are detected with a high-resolution mass spectrometer, so any impurities in them could not go unnoticed, and the clusters are ionic to begin with, so they do not need to be charged before detection.

*Actually, there exist a number of sources of large uncertainties in measured particle number concentrations. Some of them have already been pointed out by the Author in the Experimental Cluster Distribution Section (page 3):*

*"The clusters were detected using a high resolution APi-TOF (Atmospheric Pressure interface Time- Of-Flight) mass spectrometer. The largest clusters considered in the study contained one HSO4 − ion, four H2SO4 molecules and four ammonia molecules. However, it is likely that most of the clusters initially also contained some water molecules, although none were detected, and water was concluded to evaporate from the clusters inside the APi-TOF. The clusters were also assumed to lose some or all of the ammonia molecules inside the instrument prior to detection."*

*where uncertainties related to critically important hydration effect and lost of ammonia prior to detection are clearly acknowledged. These uncertainties may have a very large impact on measured particle formation rates, and, thus, some estimates of their impacts shall be included in the revised manuscript. The Author could use factors of 10 and 100 as the model "typical uncertainties" in measured particle number concentrations.*

The original comment was about impurities and charging, and these don't seem relevant in the context of the present study. As the reviewer points out, other uncertainties that are relevant are already mentioned in the manuscript. These have now been pointed out once more at the very end of the manuscript.

Page 19, lines 10-18:

[revised manuscript text omitted]

---

## Author Response (AR3)

Dear Editor,

As I already replied to the Reviewer #2, I feel that the suggestion was really strange. I did not get the data from the CLOUD community for this study, but instead took it from a previously published paper. I could simply have read the vapor and cluster concentrations from Fig. 2 of Olenius et al. (2013) if I hadn't happened to have access to the data (which obviously I did, as I was a co-author of that paper). If I thank the CLOUD community, I feel I would similarly need to thank the authors of many other papers I cite in the article. I used an MCMC algorithm published by ter Braak and Vrugt (2008) and this algorithm was crucial for my study, so shouldn't I also thank ter Braak and Vrugt in the Acknowledgements if I thank the CLOUD community? And the VODE differential equation solver by Brown et al. (1989) was an actual piece of code that was very useful, so I would certainly also need to thank Brown et al. And if I thank the co-authors of one of my earlier articles, it would seem fair to also thank the co-authors of all my other papers. And so on, and so forth. So, to keep things simple, I would prefer not to add any new acknowledgements to the paper.

Sincerely,
Oona Kupiainen-Määttä